# RETHINKING DRIVING TOPOLOGY REASONING: PLUG-AND-PLAY DISCRETE GRAPH REFINEMENT

## ABSTRACT

In autonomous driving, topology reasoning aims to recover the structured connectivity of road networks by detecting map elements and predicting their relations, providing machine-readable maps for safe and efficient operation. Surprisingly, current topology reasoning tasks do not address how to produce better discrete graphs, even though downstream modules such as planning and control rely on them. Existing methods predict continuous edge scores and then apply simple thresholding to obtain discrete graphs, but this step is neither optimized during training nor evaluated in benchmarks. As a result, it remains unclear whether their predicted continuous graphs are truly effective for downstream tasks. To bridge this gap, we propose **TopoRefine**, a universal and plug-and-play topology graph refinement module that refines continuous graphs predicted by any topology reasoning model into higher-quality discrete graphs. Specifically, it refines connectivity by learning structural patterns via a lightweight GNN-based refinement module trained in a self-supervised way. This refinement module calibrates predictions so that thresholding yields more reliable discrete structures. In addition, we are the first to introduce a discrete graph evaluation metric in this setting, the Topology Jaccard Score, tailored to directly assess the quality of discrete driving topology graph. Experiments on multiple baselines demonstrate that TopoRefine improves both continuous and discrete graph quality, making it the first framework to explicitly focus on improving discrete graph reliability in topology reasoning.

## 1 INTRODUCTION

In autonomous driving, understanding scene topology is crucial because it determines how map elements connect to form drivable routes. Topology reasoning addresses this by detecting lanes and traffic elements and predicting their connectivity, covering both lane–lane topology and lane–traffic topology. The resulting machine-readable graphs support downstream tasks such as trajectory prediction (Gu et al., 2024), path planning (Chai et al., 2019), and motion control (Hu et al., 2023), where reliable connectivity is essential for safety and efficiency.

While detection of map elements lays the foundation, the final success of topology reasoning depends on making correct connectivity predictions. Even if all map elements are detected perfectly, wrong connections can still cause failures. For example, a path planning module may choose an unsafe route if lane connections are predicted incorrectly, despite flawless detection. As shown in Figure 1b and Figure 1a, the state-of-the-art model detects traffic lights correctly but misses their lane–traffic connections, causing the planner to ignore valid passing opportunities.

Existing topology reasoning methods primarily predict a continuous topology graph by assigning confidence scores to all candidate edges (Li et al., 2023; Wu et al., 2024; Can et al., 2022; Ye et al., 2025; Lv et al., 2025). However, downstream modules such as path planning require a high-quality discrete driving topology graph, while unfortunately existing approaches do not explicitly focus on producing such reliable discrete connectivity. For clarity, we distinguish continuous from discrete topology graphs. Continuous graphs assign confidence scores to all candidate connections, producing a dense structure optimized by existing models. Discrete graphs, however, are the sparse binary connectivities required by downstream planning. Current methods convert continuous scores to discrete edges using a fixed threshold, but improvements in the continuous domain do not necessarily yield better discrete graphs. This gap motivates our focus on explicitly evaluating and refining dis-

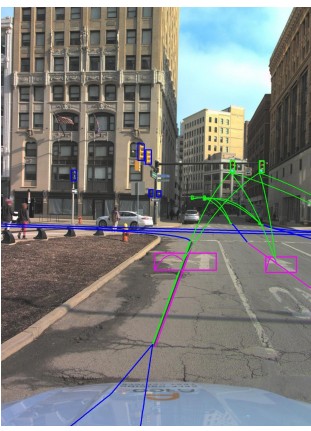 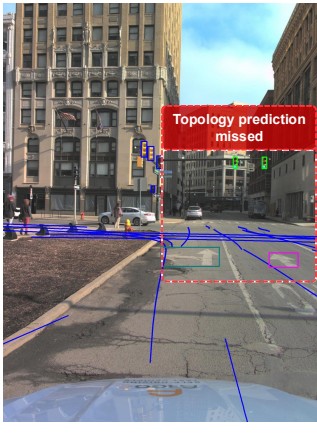 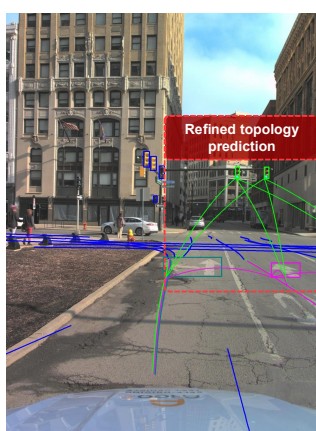

(a) Ground-Truth     (b) SMART-OL (Ye et al., 2025)     (c) SMART-OL + TopoRefine

Figure 1: The existing SOTA method (b) detect traffic lights correctly but miss their lane connections, while TopoRefine (c) can perfectly recover complete connectivity after refinement, producing discrete graphs that better match the ground truth. Green lines denote traffic light detections and their lane–traffic connections, while blue polylines show lane–lane connectivity.

crete topology. This mismatch leaves a critical gap between the focus of existing methods and the practical needs of autonomous driving. To bridge this gap, we must step back and rethink what is truly essential for enabling reliable real-world downstream tasks in autonomous driving.

From this perspective, a natural question arises: *how can continuous topology predictions be turned into better discrete graphs which can fully empower downstream tasks in autonomous driving?*

Turning this question into practice, however, is far from straightforward. The first obstacle is that existing benchmarking metrics do not evaluate discrete connectivity at all. For instance, the commonly used OpenLane-V2 score (Wang et al., 2023) measures only the quality of the continuous graph, leaving it unclear whether the resulting discrete graphs are truly effective in practice. In other words, existing metrics overlook connectivity, creating a blind spot between benchmark success and practical reliability. To close this gap, we introduce the Topology Jaccard Score (TJS), the first evaluation metric tailored to directly assess discrete connectivity. This new perspective reveals that many topology reasoning models, while achieving high scores under continuous metrics (OpenLane-V2 score), still produce poor discrete graphs, as illustrated in Figure 1b.

Yet, better evaluation metric alone is not sufficient. In order to turn continuous topology predictions into a better discrete graph, we introduce **TopoRefine**, a post-hoc and plug-and-play topology graph refinement module that bridges the gap between continuous predictions and the discrete graphs required by downstream tasks (see Figure 1c). The key idea is to refine connectivity by learning general structural patterns from data. To achieve this, TopoRefine leverages a self-supervised scheme: it perturbs graphs to generate augmented views, allowing a lightweight Graph Neural Network (GNN) to learn to distinguish real edges from fake ones. Here, the "labels" used for training refer exclusively to these augmentation-derived indicators (real vs. perturbed edges) and do *not* correspond to external factors such as city, sensor type, weather, or time-of-day, which clarifies that our method does not rely on or assume a low-label regime. These domain-transfer conditions fall outside the scope of our work and are unrelated to the type of labels used within our refinement module. This improves the separation between valid and invalid connections, so that thresholding produces more reliable discrete graphs.

More importantly, this plug-in refinement module can be seamlessly integrated into any existing topology reasoning models without retraining, because TopoRefine learns general structural patterns through self-supervised training rather than relying on model-specific designs. It is essential to emphasize that TopoRefine operates strictly as a post-hoc refinement module: it does not modify, retrain, or backpropagate gradients into the underlying topology predictor. This design choice is intentional, as our goal is to provide a model-agnostic plug-and-play component that improves the discrete structural correctness of any existing model without requiring end-to-end retraining or

architectural changes. Being lightweight and model-agnostic, it refines the output from diverse models into better discrete graphs to provide more faithful support for downstream planning and control. Together with TJS, which establishes the first metric for evaluating discrete connectivity, TopoRefine forms a complete framework for both improving and measuring the quality of discrete topology graph.

Our experimental result shows that TopoRefine improves both continuous and discrete metrics, effectively bridging confidence scores with the reliable discrete graphs required for autonomous driving. In particular, it achieves near $10\%$ relative gains on continuous topology metrics ($\text{TOP}_{ll}$, $\text{TOP}_{lt}$) and boosts discrete connectivity (TJS) by over $200\%$ on certain baselines, underscoring its substantial performance improvements across models. In summary, our key contributions are threefold:

- We are the first to explicitly highlight and address the overlooked problem of discrete graph quality in topology reasoning, emphasizing its importance for reliable downstream planning and control.

- We propose TopoRefine, a lightweight plug-and-play refinement module that can be applied to diverse topology reasoning models, improving their discrete topology graph.

- We introduce the Topology Jaccard Score, the first evaluation metric designed to assess discrete graph connectivity in driving scene topology reasoning, thereby addressing the blind spot overlooked by existing benchmarks.

## 2 RELATED WORK

### 2.1 LANE TOPOLOGY REASONING

Lane topology reasoning aims to capture connectivity among lanes, enabling scene interpretation and the definition of drivable routes. Existing methods follow several directions. Query-based approaches such as STSU (Can et al., 2022) extend DETR (Carion et al., 2020) to jointly predict lane queries and their connections. Graph-based models, including TopoNet (Li et al., 2023) and TopoMLP (Wu et al., 2024), formulate it as link prediction using scene graphs, shortest-path algorithms, or MLPs. Recent extensions like TopoLogic (Fu et al., 2024), TopoPoint (Fu et al., 2025b), and TopoFormer (Lv et al., 2025) integrate geometric priors and transformer architectures, while others such as LaneSegNet (Li et al., 2024b), SMART (Ye et al., 2025), Topo2D (Li et al., 2024a), and TopoOSMR (Zhang et al., 2024) incorporate map priors or external data (e.g., satellite imagery). Despite these advances, most works target task-specific improvements in connectivity prediction, with little attention to the quality of the resulting discrete graphs. This motivates our focus on topology graph refinement, which would help existing models produce higher-quality discrete graphs.

### 2.2 GRAPH REFINEMENT

Graph refinement adjusts node and edge features so that the resulting graph better supports downstream tasks. Prior work follows two main directions. The first learns adjacencies by imposing structural priors such as smoothness, sparsity, or connectivity directly from node signals (Dong et al., 2016; Kalofolias, 2016); for example, Franceschi et al. (2019) optimize discrete structures through bilevel learning. The second refines graphs via representation learning, where GNNs reweight or denoise edges in a post-hoc manner (Jiang et al., 2019; Zhu et al., 2021a).

Self-supervised learning has recently become a popular tool for graph refinement, as it avoids reliance on labels. Common augmentations include node perturbation, edge modification, and subgraph sampling, with objectives that reconstruct embeddings across views (Zhao et al., 2023). Representative methods include GraphCL (You et al., 2020) and GCA (Zhu et al., 2021b) (contrastive), Bootstrapped Graph Learning (Thakoor et al., 2021) (non-contrastive), and GraphMAE (Hou et al., 2022) (masked reconstruction). These approaches improve edge reliability by enforcing robustness under perturbations, making them well suited for label-scarce autonomous driving.

## 3 PROBLEM STATEMENT

**Topology Reasoning.** Given a single driving scene image frame, we represent the scene as a graph by considering two types of nodes: lane instances $\mathcal{V}_l = \{l_i \in \mathbb{R}^{11 \times 3}\}_{i=1}^{n_l}$, encoded as centerline polylines, and traffic elements $\mathcal{V}_t = \{t_j \in \mathbb{R}^d\}_{j=1}^{n_t}$, where $d$ denotes the feature dimension extracted from front-view camera (e.g., ResNet embeddings). Here, $n_l$ and $n_t$ denote the total numbers of lane and traffic element nodes, respectively. The topology reasoning task consists of detecting these entities and predicting their connectivity. Formally, we define a graph $\mathcal{G} = (\mathcal{V}, \mathcal{E})$ with $\mathcal{V} = \mathcal{V}_l \cup \mathcal{V}_t$. Connectivity is divided into two types: (1) $\mathcal{E}_{ll} \subseteq \mathcal{V}_l \times \mathcal{V}_l$, capturing lane–lane relations such as merges, splits, and successors (abbreviated as LL); and (2) $\mathcal{E}_{lt} \subseteq \mathcal{V}_l \times \mathcal{V}_t$, capturing lane–traffic relations such as lane-to-signal or lane-to-sign connections (abbreviated as LT). The resulting graph captures topological structures essential for downstream tasks such as planning and control.

In practice, a topology reasoning model $f_{\text{base}}$ outputs a predicted graph $\hat{\mathcal{G}} = (\hat{\mathcal{V}}, \hat{\mathcal{E}})$, where each edge $(u, v) \in \hat{\mathcal{E}}$ is associated with a confidence score in $[0, 1]$. A discrete graph is then obtained by thresholding, $\bar{\mathcal{E}} = \{(u, v) \in \hat{\mathcal{E}} \mid \text{score}(u, v) \geq \tau\}$. Since these confidence scores may not be fully optimized for binary connectivity, this step often results in missing or redundant edges in both LL and LT predictions.

**Topology Graph Refinement.** Topology graph refinement aims to correct the connectivity of topology graphs $\hat{\mathcal{G}}$ predicted by topology reasoning model $f_{\text{base}}$. Given map element nodes $\hat{\mathcal{V}}$ detected by $f_{\text{base}}$, a refinement module $f_\theta$ outputs a refined graph $\tilde{\mathcal{G}} = (\hat{\mathcal{V}}, \tilde{\mathcal{E}})$, adjusting lane–lane (LL) and lane–traffic (LT) relations. The goal is to reduce errors such as missing successors, redundant merges, and incorrect lane-to-signal links, so that $\tilde{\mathcal{E}}$ better matches the ground-truth topology $\mathcal{E}^*$. Refined edges are fused with original edge predictions $\hat{\mathcal{E}}$, yielding discrete graphs that more accurately reflect real-world road connectivity for downstream planning and control.

## 4 TOPOREFINE

We propose TopoRefine, a universal and lightweight topology refinement module that improves continuous connectivity by learning structural patterns from large-scale augmented data, thereby producing higher-quality discrete graphs required by downstream tasks. Designed as a plug-and-play component, it can be applied post-hoc to any topology reasoning model without retraining.

### 4.1 SELF-SUPERVISED TOPOLOGY GRAPH REFINEMENT

TopoRefine refines continuous topology predictions into reliable discrete graphs through a self-supervised framework with three components. Graph augmentation generates perturbed and negative samples to provide label-free supervision and robustness to noise. A lightweight GNN refinement model then learns structural patterns to predict edge confidence. Finally, an adaptive refinement loss handles class imbalance across relation types, enabling stable training. Together, these components yield faithful discrete graphs that better support downstream tasks.

**Graph Augmentation.** We construct augmented graphs from ground-truth annotations to provide label-free training signals (Fig. 2a), enabling a single refinement model to generalize across different topology reasoning models. Following a self-supervised paradigm, we first add isolated nodes as negative nodes while keeping the original topology fixed, and then perturb nodes to simulate the potential deviation from predictions to ground-truth.

Specifically, we first expand the node set with perturbed copies of annotated nodes, referred to as fake nodes. These automatically form negative edges (label 0) with existing nodes, ensuring that the augmented graph preserves the same output dimensionality as topology reasoning predictions $\hat{\mathcal{G}}$ (e.g., $|\mathcal{E}_{ll}^+| = |\hat{\mathcal{E}}_{ll}|$, $|\mathcal{E}_{lt}^+| = |\hat{\mathcal{E}}_{lt}|$). In terms of node feature, each lane node $l_i \in \mathcal{V}_l^+$ has polyline features $\mathbf{x}_{l,i}$, while each traffic element $t_j \in \mathcal{V}_t^+$ has visual features $\mathbf{x}_{t,j}$ extracted based on its bounding box $\mathbf{b}_j$. We perturb these features by adding Gaussian noise:

$$\mathbf{z}' = \mathbf{z} + \epsilon, \quad \epsilon \sim \mathcal{N}(0, \sigma^2 I), \quad \mathbf{z} \in \{\mathbf{x}_{l,i}, \mathbf{b}_j\}. \tag{1}$$

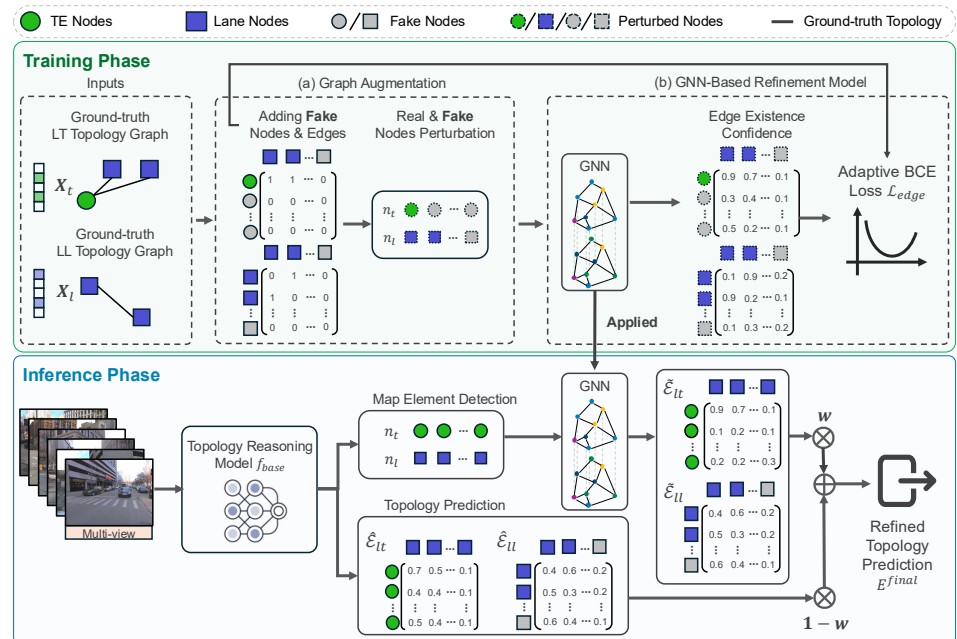

Figure 2: Overall framework of TopoRefine. **Training phase:** (a) Graph augmentation: lane nodes $\mathbf{X}_l$ (centerline polylines) and traffic element features $\mathbf{X}_t$ (embeddings from front-view detections) are expanded with fake nodes and perturbed to generate augmented views. Note that the supervision used here is entirely generated from perturbation-based augmentation and is applied only to train the refinement module itself. The base topology prediction network remains frozen throughout. This decoupled design ensures that TopoRefine functions as a post-training refinement step rather than an end-to-end optimization framework, making it compatible with diverse existing models without altering their training pipelines. (b) GNN-based refinement: a lightweight GNN processes the augmented graph and predicts edge confidences, optimized with an adaptive BCE loss. **Inference phase:** given topology predictions $\hat{E}_{ll}, \hat{E}_{lt}$ from a baseline model $f_{\text{base}}$, the refinement module produces $\tilde{E}_{ll}, \tilde{E}_{lt}$, which are fused with $\hat{E}_{ll}, \hat{E}_{lt}$ to yield refined discrete graphs for downstream planning and control.

It is worth mentioning that, the node perturbation is applied on both real nodes and fake nodes, with a key motivation unique in our task. Unlike the existing graph augmentation approaches designed for general domain (You et al., 2020; Hou et al., 2022), there exists a gap between our training data augmented from ground-truth graph and inference data predicted by the topology reasoning model $f_{base}$. If real nodes were kept unchanged, as in prior approaches, the refinement model would fail to generalize well to these imperfect predictions. To address this, we apply only mild perturbations to real nodes, ensuring they remain within distance thresholds $\delta_l, \delta_t$ of the ground truth while maintaining a clear distinction from fake nodes (see Appendix A.2). During training, the perturbation magnitude decreases adaptively for fake nodes and increases adaptively for real nodes, inspired by the principle of active learning (Settles, 2009). In one hand, this makes the refinement module robust to imperfect detections during inference, which are unavoidable in practice. In another hand, fake nodes perturbed beyond thresholds serve as hard negatives, strengthening the force of contrastive learning. Together, this dual strategy prevents overfitting to clean annotations and enables the model to generalize better to noisy real-world inputs.

**GNN-Based Refinement Model.** Given augmented graph $\mathcal{G}^+$, refinement is formulated as edge confidence prediction: the goal is to estimate the likelihood of edges so that thresholding yield reliable discrete topology graph (see 2(b)). GNNs are well suited for this goal because they capture relational structure, allowing each node to update its embedding based on neighbors and learn general connection rules for driving topology graph. Formally, node embeddings are updated as,

$$\mathbf{h}_v^{(k)} = \phi\left(\mathbf{h}_v^{(k-1)}, \square_{u \in \mathcal{N}(v)} \psi(\mathbf{h}_v^{(k-1)}, \mathbf{h}_u^{(k-1)})\right), \tag{2}$$

where $\psi$ is the message function, $\square$ an aggregator, and $\phi$ an update function. Specifically, a two-layer heterogeneous Graph Attention Network (GAT) is employed to predict edge confidence after comparing different GNN model architectures (Appendix A.5):

$$\mathbf{h}_v = f_\theta(v, \mathcal{G}^+), \quad \tilde{e}_{uv} = \mathbf{h}_u^\top \mathbf{h}_v / (\|\mathbf{h}_u\| \|\mathbf{h}_v\|), \quad v \in \mathcal{V}^+. \tag{3}$$

**Adaptive Refinement Loss.** Ground-truth topology is highly sparse: only a small fraction of lane–lane and lane–traffic connections are valid, and adding fake nodes further increases the imbalance between positive edges and negative edges. To handle this, we introduce an adaptive BCE loss that normalizes within each relation type and balances their scales. Given relation $r \in \{ll, lt\}$ with positives $E_r^{pos}$ and negatives $E_r^{neg}$, our loss function $\mathcal{L}_{\text{edge}}$ is formulated as:

$$\mathcal{L}_{\text{BCE}}^{(r)} = -\frac{1}{|E_r^{\text{pos}}| + |E_r^{\text{neg}}|} \left( \sum_{(u,v) \in E_r^{\text{pos}}} \log \hat{e}_{uv} + \sum_{(u,v) \in E_r^{\text{neg}}} \log(1 - \hat{e}_{uv}) \right), \tag{4}$$

$$\mathcal{L}_{\text{edge}} = \frac{\mathcal{L}_{\text{BCE}}^{(ll)}}{|\mathcal{E}_{ll}|} + \frac{\mathcal{L}_{\text{BCE}}^{(lt)}}{|\mathcal{E}_{lt}|}, \tag{5}$$

where normalization avoids negatives dominating positives, and scaling ensures equal weight across different relations, yielding stable training under sparse augmented graphs.

**Edge Confidence Calibration.** For every possible edge $(u, v)$, we calibrate the edge existence confidence by fusing the edge confidence $\hat{e}_{uv}$ predicted by the topology reasoning model $f_{base}$ with the refined score $\tilde{e}_{uv}$ from our module (see Figure 2(c)). The calibrated confidence is computed as,

$$e_{uv}^{\text{final}} = w_r \cdot \tilde{e}_{uv} + (1 - w_r) \cdot \hat{e}_{uv}, \qquad r \in \{ll, lt\}, \tag{6}$$

$w_r$ is a relation-specific fusion weight, with larger value indicating greater reliance on our module. This calibration sharpens noisy continuous scores into more reliable discrete connectivity.

## 4.2 Discrete Graph Evaluation

In the existing literature, the OpenLane-V2 TOP score (Wang et al., 2023) is the most widely used topology metric. It computes mean average precision over predicted edge confidences. However, a key limitation is that it evaluates only the continuous graph before thresholding, whereas downstream modules operate on discrete graphs obtained after thresholding. As a result, a model can achieve a high TOP score yet still produce missing or spurious connections in real-world applications. This mismatch highlights a long-overlooked need for a metric that explicitly evaluates the quality of discrete graphs.

A natural candidate for discrete graph evaluation from graph theory is the graph edit distance (GED) (Gao et al., 2010), which measures the minimum number of node and edge edits required to transform one graph into another. GED accounts for both detection and connectivity errors, treating them as equally costly. However, for our purposes, GED is not ideal: we aim to specifically evaluate the quality of edge predictions, and GED node-edit operations are unnecessary. Moreover, exact GED computation is NP-hard, with exponential complexity on the order of $\mathcal{O}(|\mathcal{V}|^N)$, making it impractical for large-scale driving graphs.

**Topology Jaccard Score (TJS).** To address these issues, we introduce the Topology Jaccard Score (TJS), a metric that is both efficient and tailored to discrete connectivity evaluation in topology reasoning. TJS represents a graph as an adjacency list and reduces graph comparison to a set comparison problem. Inspired by Jaccard Similarity (Jaccard, 1912), it measures the overlap between predicted and ground-truth edges, normalized by their union. The key challenge is that edge matching requires detection-aware node correspondence. Let $\mathcal{E}^*$ be the ground-truth edges and $\bar{\mathcal{E}}_\tau = \{(i, j) \in \hat{\mathbf{A}} : \hat{p}_{ij} > \tau\}$ be the predicted edges thresholded at $\tau = 0.5$. Following the OpenLane-V2 (Wang et al., 2023), each ground-truth node $v \in \mathcal{V}^*$ is matched to the highest-confidence detection $\hat{v} \in \hat{\mathcal{V}}$ within distance threshold $\delta_l, \delta_t$ of $v$, with unmatched detections counted as false positives. The set of true-positive detections is $\mathcal{V}_{\text{TP}} = \{\hat{v} \in \hat{\mathcal{V}} : \exists v \in \mathcal{V}^* \text{ s.t. } \hat{v} = \arg\max_{\hat{u} \in C(v)} \hat{p}(\hat{u})\}$, where

$C(v)$ is the set of candidate detections near $v$. Formally, the Topology Jaccard Score is evaluated as:

$$\text{TJS}(\mathcal{E}^*, \hat{\mathcal{E}}_\tau) = \frac{|\{(u,v) \in \bar{\mathcal{E}}_\tau \cap \mathcal{E}^*: \ u, v \in \mathcal{V}_{\text{TP}}\}|}{|\mathcal{E}^* \cup \bar{\mathcal{E}}_\tau|}, \tag{7}$$

where the numerator counts correctly predicted edges between matched true positive detections, and the denominator includes all ground-truth and high-confidence predicted edges. TJS runs in linear time $\mathcal{O}(|\mathcal{E}|)$ and provides a detection-aware measure of discrete connectivity that directly complements continuous graph metrics such as the OpenLane-V2 TOP score.

## 5 EXPERIMENTS

### 5.1 DATASET AND METRICS

**Dataset.** We conduct experiments on the OpenLane-V2 benchmark (Wang et al., 2023), a large-scale dataset for perception and reasoning in autonomous driving. All experiments are performed on Subset A, derived from Argoverse 2 (Wilson et al., 2023), which contains 1,000 scenes with multi-view images and annotations at $2\,\text{Hz}$. Lane centerlines are given as ordered 3D polylines of 201 points within a spatial range of $[-50, 50]\,\text{m}$ longitudinally and $[-25, 25]\,\text{m}$ laterally. We downsample them into 11 points, following the standard schema in topology reasoning models Li et al. (2023), and use these as lane node features $\mathbf{X}_l$. About 90% of frames contain more than 10 centerlines, with some exceeding 40. Traffic elements are annotated as 2D bounding boxes in front-view images and span 13 semantic categories (e.g., traffic light states, direction signals). Each lane typically has one predecessor and one successor, with up to seven outgoing connections in complex intersections (Li et al., 2023; Wu et al., 2024). Following existing topology reasoning models, detections are capped at $n_l = 200$ lanes and $n_t = 100$ traffic elements per frame.

**Evaluation Metrics.** We evaluate topology reasoning with the OpenLane-V2 (v2.1.0) TOP score (Wang et al., 2023). For each predicted edge, a confidence score is produced, and edges are ranked accordingly. Predicted edges are then matched to ground-truth edges under geometric thresholds, and mean average precision is computed. Results are reported separately for lane–lane connectivity ($\text{TOP}_{ll}$) and lane–traffic connectivity ($\text{TOP}_{lt}$), Appendix A.7 provides more details on its calculation.

To go beyond this, we report two complementary measures. First, TJS (Eq. 7) directly evaluates the quality of binarized discrete graphs by measuring overlap between predicted and ground-truth edge sets. Second, we compute the margin to the upper bound. The upper bound $\text{UB}_r$ ($r \in \{ll, lt\}$) assumes perfect topology prediction on detected nodes, where all true positives are assigned confidence 1.0 and connected exactly as in the ground truth, while false positives and missed detections remain unchanged. The $\text{margin}_r = \text{UB}_r - \text{TOP}_r$ therefore reflects how much performance is still left to close under continuous metrics, conditioned on imperfect detections. Overall, we report $\text{TOP}_{ll} \uparrow$, $\text{TOP}_{lt} \uparrow$, JS $\uparrow$ for discrete evaluation, and $\text{margin}_{ll} \downarrow$, $\text{margin}_{lt} \downarrow$ as a complementary view of continuous performance relative to its detection-conditioned upper bound.

### 5.2 IMPLEMENTATION DETAILS

**Baselines and Setup.** We evaluate five topology reasoning models with public code and available checkpoints: TopoNet, TopoMLP, SMART-OL (TopoMLP), Topo2D, and TopoLogic (Table 1). For each baseline, we use outputs from the released checkpoints as inputs to TopoRefine; models without checkpoints are not included. All experiments are run on a single NVIDIA H200 GPU using the same refinement model across baselines, demonstrating the plug-and-play nature of our approach.

**Feature Extraction and Training.** We initialize node features with DinoV2-ViT-L embeddings (Oquab et al., 2023), which provide 1024-dimensional representations of front-view images. To study encoder choice, we also test DinoV3 (Siméoni et al., 2025) and ResNet-50 (He et al., 2016), with results reported in the ablation study. TopoRefine is trained for 200 epochs with batch size 64 using AdamW (Loshchilov & Hutter, 2017) (lr = 0.001, weight decay = 0.01) and CosineAnneal-ingLR (Loshchilov & Hutter, 2016), decaying to $10^{-4}$. Training TopoRefine takes about 1.5 hours and validation about 45 minutes on a single H200 GPU on Subset A. Unlike topology reasoning models such as SMART (Ye et al., 2025) and TopoNet (Li et al., 2023), TopoRefine does not train

Table 1: Comparison of methods on the OpenLane-V2 Subset A dataset using OpenLane-V2 metrics. Best results are shown in bold and the second-best are underlined. Percentage changes indicate relative improvements over the corresponding baseline before adding TopoRefine.

| Input type | Method | Venue | $TOP_{ll}$ ↑ | $TOP_{lt}$ ↑ |
|---|---|---|---|---|
| Perspective images | STSU (Can et al., 2022) | ICCV 2021 | 2.9 | 19.8 |
| | VectorMapNet (Liu et al., 2023) | ICML 2023 | 2.7 | 9.2 |
| | MapTR (Liao et al., 2022) | ICLR 2023 | 5.9 | 15.1 |
| | TopoNet (Li et al., 2023) | Arxiv 2023 | 10.9 | 23.8 |
| | TopoMLP (Wu et al., 2024) | ICLR 2024 | 21.6 | 26.9 |
| | Topo2D (Li et al., 2024a) | Arxiv 2024 | 22.3 | 26.2 |
| | RoadPainter (Ma et al., 2024) | ECCV 2024 | 22.8 | 27.2 |
| | TopoFormer (Lv et al., 2025) | CVPR 2025 | 24.1 | 29.5 |
| | TopoPoint (Fu et al., 2025a) | Arxiv 2025 | 28.7 | 30.0 |
| Perspective images + SD maps | TopoOSMR (Zhang et al., 2024) | IROS 2024 | 17.1 | 26.8 |
| | SMERF (Luo et al., 2024) | ICRA 2024 | 15.4 | 25.4 |
| | TopoLogic (Fu et al., 2024) | NeurIPS 2024 | 23.9 | 25.4 |
| | RoadPainter (Ma et al., 2024) | ECCV 2024 | 29.6 | 29.5 |
| Perspective images + Map priors | SMART (TopoNet) (Ye et al., 2025) | ICRA 2025 | 27.5 | 33.1 |
| | SMART (TopoMLP) (Ye et al., 2025) | ICRA 2025 | 37.0 | 33.0 |
| Perspective images | TopoNet + TopoRefine | Ours | $21.8_{\uparrow 100\%}$ | $25.8_{\uparrow 19.7\%}$ |
| | TopoMLP + TopoRefine | | $24.4_{\uparrow 12.3\%}$ | $28.7_{\uparrow 6.5\%}$ |
| | Topo2D + TopoRefine | | $24.3_{\uparrow 9.2\%}$ | $27.3_{\uparrow 4.3\%}$ |
| | TopoLogic + TopoRefine | | $24.5_{\uparrow 2.1\%}$ | $27.2_{\uparrow 2.1\%}$ |
| | SMART (TopoMLP) + TopoRefine | | $\mathbf{40.1}_{\uparrow 8.5\%}$ | $\mathbf{35.4}_{\uparrow 7.3\%}$ |

a full model from scratch; it is a lightweight refinement module that adds only a small amount of extra computation on top of existing models.

**Graph Augmentation and Refinement.** We generate augmented graphs by perturbing node features with Gaussian noise ($\sigma_b = 13$, $\sigma_p = 0.07$). If sampled noise falls outside the valid range, the standard deviations are adaptively rescaled (multiplied by 0.8 or 1.2) so that new nodes either remain plausible (valid) or clearly serve as fake negatives, while preserving structural consistency. Refinement is performed with a two-layer GATv2 encoder (Brody et al., 2021) in PyG (Fey & Lenssen, 2019), using hidden size 64, ReLU, and dropout 0.1. Lane and traffic features ($\mathbf{X}_l$, $\mathbf{X}_t$) are first projected into a shared space by MLP heads and then encoded by GAT. Edge scores are computed with a dot-product decoder like Eq. 3.

## 5.3 MAIN RESULTS

**OpenLane-V2 Score Evaluation.** Table 1 summarizes results on the Subset A benchmark. For methods without public checkpoints, we follow the numbers reported in their original papers; for released models, we apply TopoRefine directly on the provided pretrained weights. The table also specifies the input modality of each method, adopted from Ye et al. (2025). All results are obtained with the same refinement checkpoint, showing the universal, plug-and-play nature of TopoRefine.

Our refinement leaves detection performance unchanged but consistently improves topology reasoning by re-estimating edge confidences with a self-supervised GNN. Discrete graphs better align with road topology, with especially strong gains on lane–lane relations ($TOP_{ll}$). Weaker baselines such as TopoNet nearly double their lane–lane performance ($10.9 \rightarrow 21.8$), while stronger models like SMART (TopoMLP) also benefit ($+8.5\%$ $TOP_{ll}$, $+7.3\%$ $TOP_{lt}$). Other architectures, including TopoMLP and Topo2D, achieve consistent $+9–12\%$ improvements on $TOP_{ll}$ and $+4–7\%$ on $TOP_{lt}$.

**Gap-to-Upper-Bound Evaluation.** We next examine how much TopoRefine narrows the gap to the detection-conditioned upper bound defined in Section 5.1. Table 2 shows that margins are consistently reduced across baselines: TopoNet shrinks $margin_{ll}$ from 13.4 to 2.5 (↓ 81.3%), while the strong SMART (TopoMLP) baseline still achieves an $80.1\%$ reduction. Lane–traffic margins improve more modestly (30–60%) but remain consistently better. Overall, TopoRefine pushes predictions substantially closer to their theoretical best, with especially large gains in lane–lane connectivity where structural consistency is most critical.

**Discrete Graph Evaluation.** Table 2 reports discrete graph quality using TJS. TopoRefine delivers substantial gains across all baselines: lane–lane JS improves by over $100\%$ for TopoNet, $220\%$ for

Table 2: Combined evaluation across baselines. We report the detection-conditioned upper bound (UB), the gap to this bound (margin; smaller is better), and discrete graph quality measured by TJS (larger is better). "After" results include green subscripts showing relative improvement.

| Method | UB (TOP) | | $\text{margin}_{ll}\downarrow$ | | $\text{margin}_{lt}\downarrow$ | | $\text{TJS}_{ll}$ (%) ↑ | | $\text{TJS}_{lt}$ (%) ↑ | |
|---|---|---|---|---|---|---|---|---|---|---|
| | $\text{TOP}_{ll}$ | $\text{TOP}_{lt}$ | Bef. | Aft. | Bef. | Aft. | Bef. | Aft. | Bef. | Aft. |
| TopoNet | 24.4 | 28.8 | 13.4 | $2.5_{\downarrow81.3\%}$ | 7.3 | $3.0_{\downarrow58.5\%}$ | 16.3 | $32.9_{\uparrow102.0\%}$ | 31.0 | $55.8_{\uparrow79.8\%}$ |
| TopoMLP | 25.0 | 31.4 | 3.3 | $0.7_{\downarrow80.4\%}$ | 4.4 | $2.7_{\downarrow39.6\%}$ | 18.5 | $59.7_{\uparrow222.9\%}$ | 31.5 | $59.8_{\uparrow89.8\%}$ |
| Topo2D | 25.0 | 29.8 | 2.8 | $0.7_{\downarrow74.3\%}$ | 3.6 | $2.5_{\downarrow31.4\%}$ | 18.9 | $55.0_{\uparrow190.6\%}$ | 34.5 | $51.5_{\uparrow49.0\%}$ |
| TopoLogic | 26.0 | 30.6 | 2.0 | $1.5_{\downarrow24.8\%}$ | 5.3 | $3.3_{\downarrow36.7\%}$ | 21.0 | $43.9_{\uparrow109.4\%}$ | 32.5 | $53.3_{\uparrow64.3\%}$ |
| SMART (TopoMLP) | 40.9 | 38.9 | 3.9 | $0.8_{\downarrow80.1\%}$ | 5.9 | $3.5_{\downarrow41.2\%}$ | 38.1 | $87.7_{\uparrow130.0\%}$ | 36.6 | $82.8_{\uparrow126.4\%}$ |

TopoMLP, and 130% even for SMART (TopoMLP), while lane–traffic JS improves by 50–126%. Compared to margin-based metrics, which mainly capture edge-level classification against an upper bound, JS directly measures overlap between predicted and ground-truth graphs and reveals far more significant improvements. This shows that TopoRefine both narrows the gap to the theoretical best and yields discrete graphs that more faithfully match real-world road topology.

**Qualitative Comparison.** Figure 3 highlights how refinement improves discrete topology. In both TopoNet and TopoMLP, baseline predictions leave critical connections missing or fragmented, such as unlinked traffic lights or inconsistent lane merges. After applying TopoRefine, these gaps are consistently corrected: traffic signals are properly attached to lanes, lane centerlines align more faithfully with the ground truth, and the overall graph becomes structurally coherent. This qualitative evidence reinforces the quantitative results, showing that our refinement yields more usable topological maps for downstream driving tasks. Check more qualitative results in Appendix A.3.

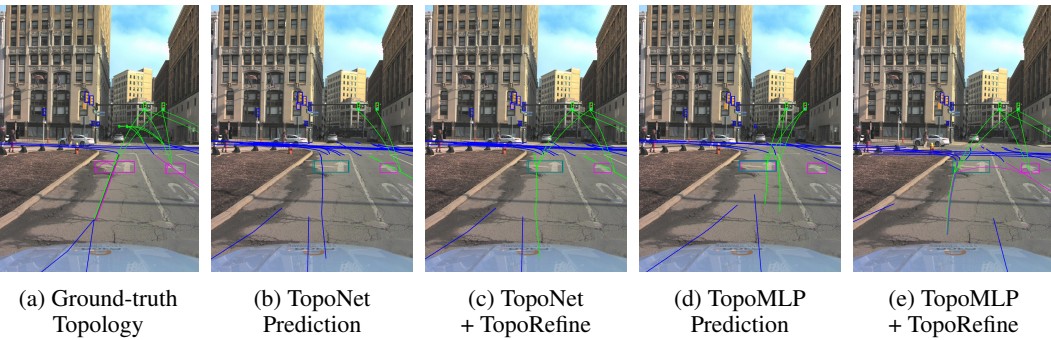

| (a) Ground-truth Topology | (b) TopoNet Prediction | (c) TopoNet + TopoRefine | (d) TopoMLP Prediction | (e) TopoMLP + TopoRefine |
|---|---|---|---|---|

Figure 3: Qualitative comparison of topology predictions before and after refinement.

## 5.4 ABLATION STUDY

We conduct ablation studies to examine the effect of feature extractors (Appendix A.4), loss functions, and real-node perturbation. All experiments are performed under the same training setup, using TopoNet as the baseline topology reasoning model for refinement.

Table 3: Ablation studies on loss functions and real-node perturbation. Results are reported on lane–lane ($\text{TOP}_{ll}$) and lane–traffic ($\text{TOP}_{lt}$) topology.

**(a) Loss Functions**

| Loss | $\text{TOP}_{ll}$ | $\text{TOP}_{lt}$ |
|---|---|---|
| BCE | 12.1 | 22.0 |
| Hybrid BCE–Focal | **22.2** | 22.0 |
| Adaptive BCE (Ours) | 21.8 | **25.8** |

**(c) Real-node Perturbation**

| Strategy | $\text{TOP}_{ll}$ | $\text{TOP}_{lt}$ |
|---|---|---|
| w/o Perturbation | 21.7 | 25.4 |
| w/ Perturbation | **21.8** | **25.8** |

**Loss Function.** We compare our adaptive BCE loss against standard BCE and a Hybrid BCE–Focal variant. For BCE, we set the positive weight to 0.2. For Hybrid BCE–Focal, we set $\alpha = 0.75$, $\gamma = 2.0$, and also define the final loss as $\mathcal{L}_{\text{Hybrid}} = 0.9 \cdot \mathcal{L}_{\text{BCE}} + 0.1 \cdot \mathcal{L}_{\text{Focal}}$, favoring BCE while leveraging focal loss at the same time. Table 3(a) shows that adaptive BCE achieves the best overall

performance. Unlike commonly used BCE and Hybrid BCE–Focal, it requires no extra hyperparameters, making it a more robust choice for graph refinement where graph sizes and sparsity vary.

**Real Node Perturbation.** Prior work in graph self-supervised learning has shown that node-level perturbations are an effective augmentation strategy for improving robustness (Zhu et al., 2021b; 2020). In our framework, real-node perturbation plays a limited but conceptually important role: without introducing any variation to real nodes, the refinement module may overfit to the exact annotated geometry rather than learning structural connectivity patterns that generalize to unseen scenes. Adding small, semantics-preserving perturbations helps regularize the model by encouraging it to focus on relational structure instead of memorizing raw coordinates. We clarify that Table 3(b) isolates only the real-node perturbation component. This perturbation is a small auxiliary part of our full framework—which also includes self-supervised GNN training, fake-node perturbation for negative samples, and the adaptive BCE loss—so its marginal improvement is expected and does not reflect the contribution of the overall method. As shown in the ablation study, real-node perturbation provides small but consistent improvements in topology metrics, aligning with its intended role as a lightweight augmentation rather than a primary performance driver.

## 6 CONCLUSION

We revisit topology reasoning through the perspective of discrete graph quality, exposing the gap between continuous edge scores and the discrete connectivity required by downstream tasks in autonomous driving. To address this, we introduce TopoRefine, a universal and plug-and-play topology refinement module that post-hoc improves any topology reasoning models through self-supervised augmentations and a lightweight heterogeneous GNN. Experiments on OpenLane-V2 baselines show that TopoRefine consistently improves TOP scores, substantially narrows gap-to-upper-bound margins, and delivers significant gains in discrete evaluation (e.g., TJS). These results demonstrate that refining edge confidences prior to thresholding offers a simple and effective approach to obtain reliable road graphs. Our work establishes discrete-graph evaluation as a core objective for topology reasoning and provides a practical framework to better align predictions with the requirements of downstream autonomous driving tasks.

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

# A APPENDIX

## CONTENTS OF APPENDIX

## A.1 DISCLOSURE OF LLM USE

We used a large language model (OpenAI's ChatGPT) to assist with paper writing. Specifically, the LLM was used for drafting text, revising passages, and changing language style. All outputs from the LLM were carefully reviewed and revised by the authors before inclusion in the main text. The authors take full responsibility for the accuracy, originality, and validity of all content in this submission.

## A.2 ADAPTIVE VARIANCE OF NODE PERTURBATION

Nodes are labeled as real or fake based on geometric consistency with the ground truth after perturbation. We denote by $\delta_l$ the distance threshold for lane nodes and by $\delta_t$ the distance threshold for traffic-element nodes, aligned with the OpenLane-V2 evaluation settings (Wang et al., 2023).

For lane polylines, distance is measured by the Chamfer distance between ground-truth polyline $\mathbf{p}$ and perturbed polyline $\hat{\mathbf{p}}$:

$$d_{\text{Chamfer}}(\mathbf{p}, \hat{\mathbf{p}}) = \frac{1}{|\mathbf{p}|} \sum_{x \in \mathbf{p}} \min_{y \in \hat{\mathbf{p}}} \|x - y\|_2 + \frac{1}{|\hat{\mathbf{p}}|} \sum_{y \in \hat{\mathbf{p}}} \min_{x \in \mathbf{p}} \|x - y\|_2. \tag{8}$$

A perturbed lane node is considered real if

$$d_{\text{Chamfer}}(\mathbf{p}, \hat{\mathbf{p}}) \leq \delta_l, \quad \delta_l = 3.0. \tag{9}$$

For traffic elements, distance is measured by Intersection-over-Union (IoU) between ground-truth bounding box $\mathbf{b}$ and perturbed box $\hat{\mathbf{b}}$:

$$\text{IoU}(\mathbf{b}, \hat{\mathbf{b}}) = \frac{|\mathbf{b} \cap \hat{\mathbf{b}}|}{|\mathbf{b} \cup \hat{\mathbf{b}}|}. \tag{10}$$

A perturbed traffic-element node is considered real if

$$\text{IoU}(\mathbf{b}, \hat{\mathbf{b}}) \geq \delta_t, \quad \delta_t = 0.75. \tag{11}$$

Perturbations are applied iteratively with variance $\sigma$ (Eq. 1). At each iteration, the consistency of a perturbed node is checked against its threshold ($\delta_l$ for lanes, $\delta_t$ for traffic elements). If the condition is satisfied, the node is labeled as real and the variance is reduced ($\sigma \leftarrow 0.8\sigma$); if not, the node is labeled as fake and the variance is increased ($\sigma \leftarrow 1.2\sigma$). This adaptive scheme continues until the threshold condition is enforced, ensuring a clear separation between real and fake nodes while respecting OpenLane-V2 tolerances.

## A.3 MORE QUALITATIVE RESULTS

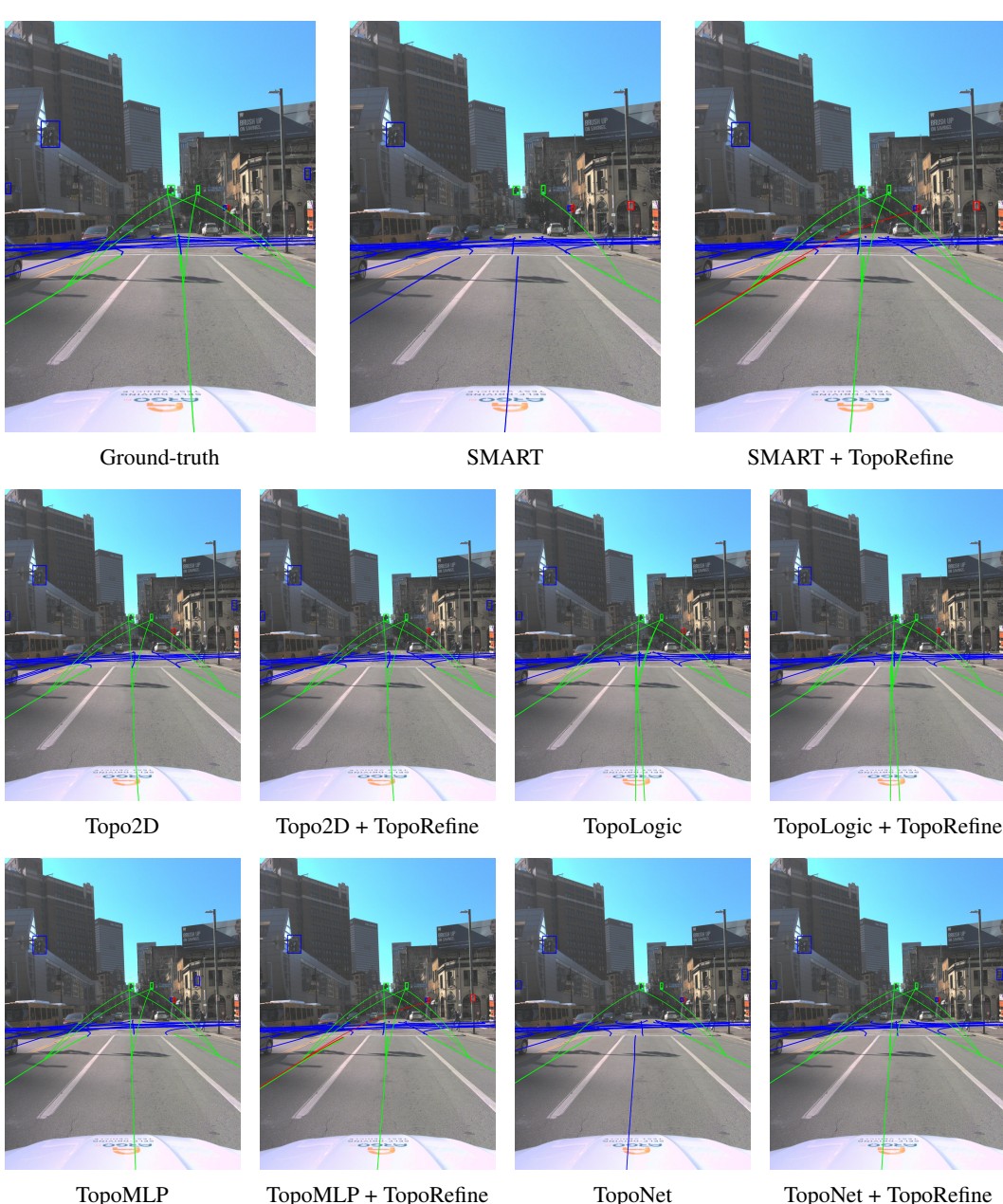

Figure 4: Qualitative results for scene 10023: predictions before and after refinement with TopoRefine across baselines.

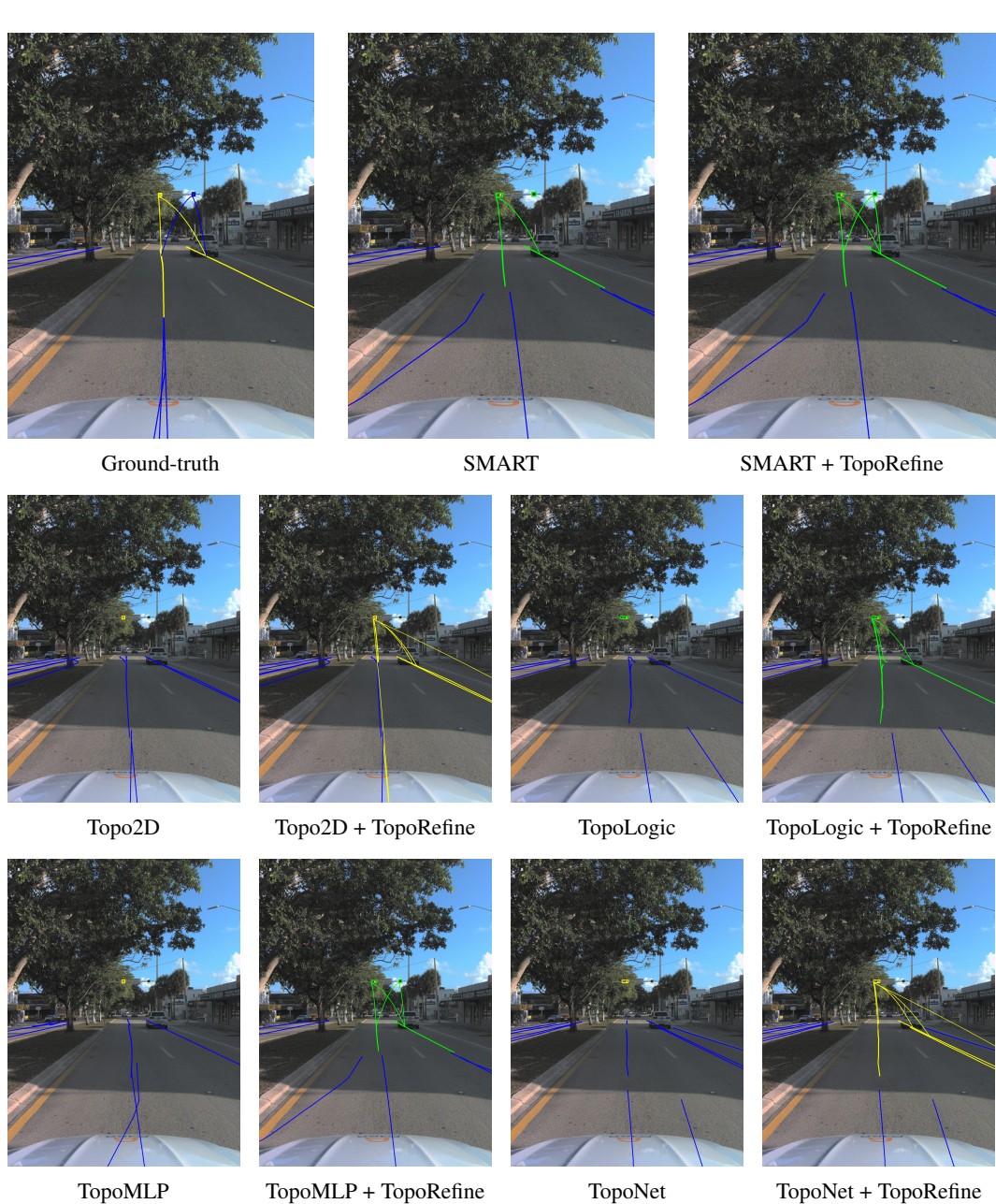

Figure 5: Qualitative results for scene 10013: predictions before and after refinement with TopoRefine across baselines (SMART, Topo2D, TopoLogic, TopoMLP, TopoNet). All panels correspond to the same ground-truth view (left).

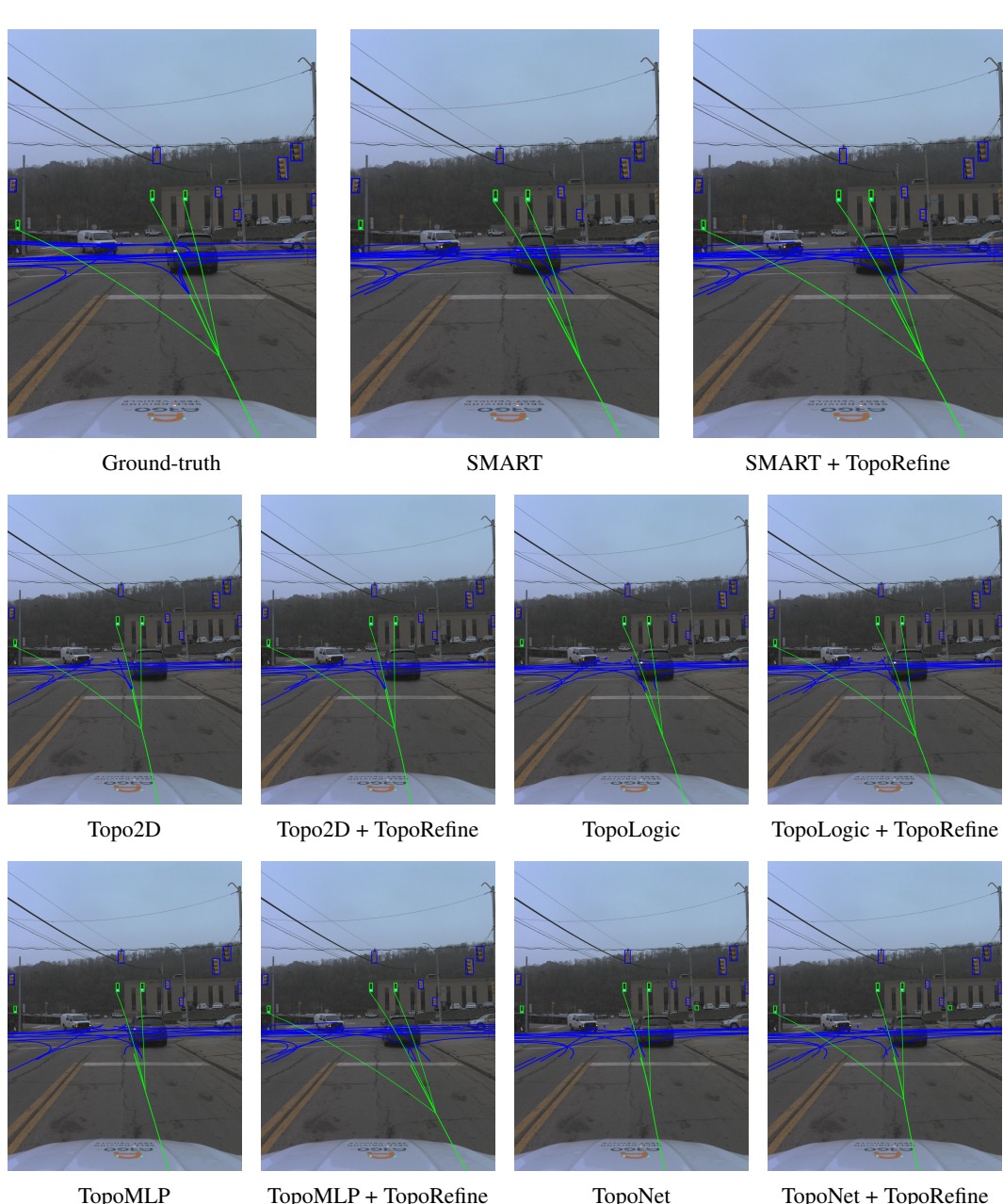

Figure 6: Qualitative results for scene 10021: predictions before and after refinement with TopoRefine across baselines (SMART, Topo2D, TopoLogic, TopoMLP, TopoNet). All panels correspond to the same ground-truth view (left).

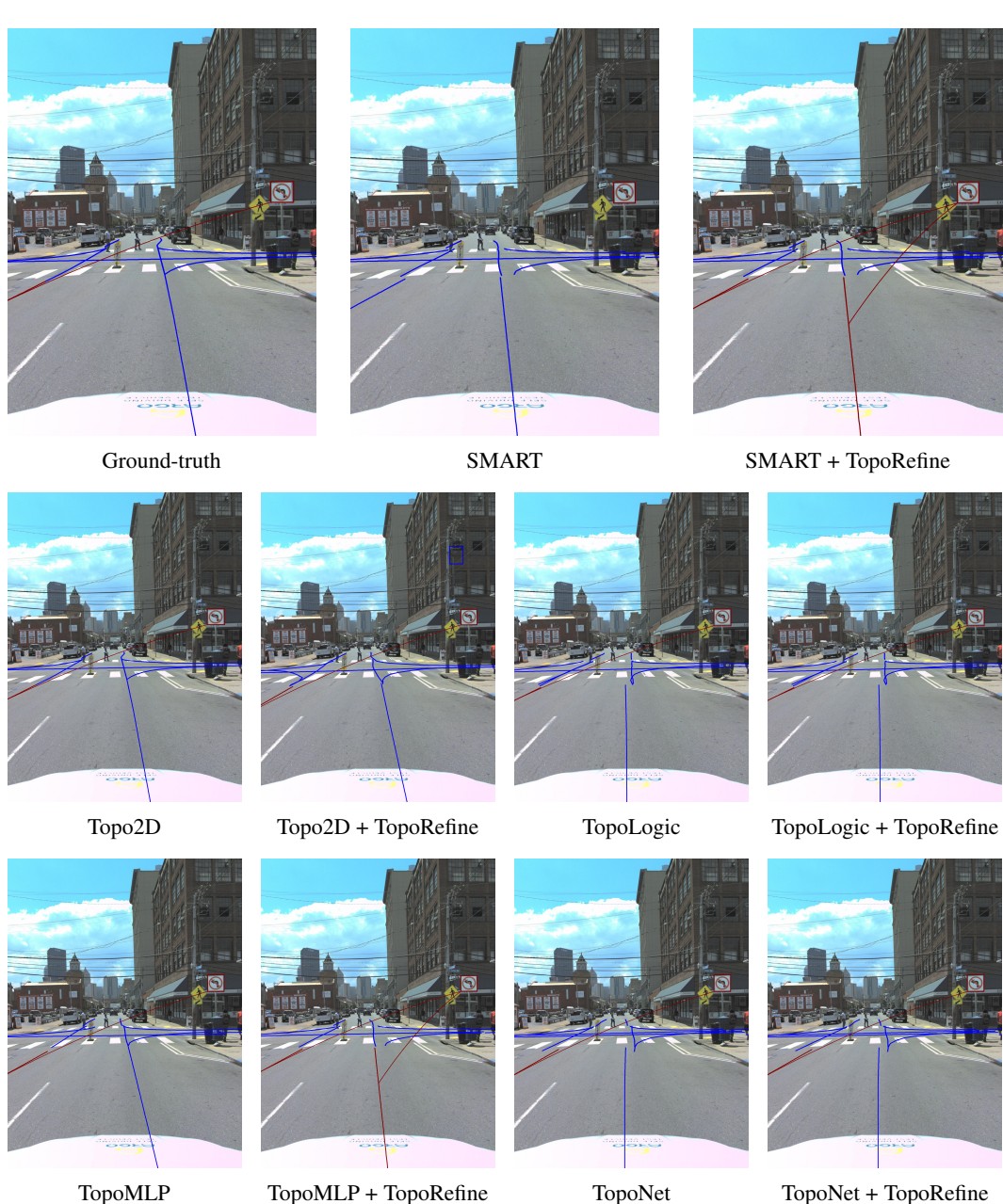

Figure 7: Qualitative results for scene 10030: predictions before and after refinement with TopoRefine across baselines.

## A.4 ABLATION ON FEATURE EXTRACTORS

We compare feature extraction for traffic elements $\mathbf{X}_t$ from DINOv2, DINOv3, and ResNet-50. For ResNet-50, we directly use the pretrained embeddings provided by TopoNet (Li et al., 2023), while for DINOv2 and DINOv3 we retrain the refinement module with their respective features. Shown in Table 4, all models achieve similar performance, with DINOv2-ViT-L slightly outperforming the others, indicating that our refinement module is robust to different feature extraction models.

Table 4: Ablation study on feature extractors.

| Extractor | Dim | $\text{TOP}_{ll}$ | $\text{TOP}_{lt}$ | Extractor | Dim | $\text{TOP}_{ll}$ | $\text{TOP}_{lt}$ |
|---|---|---|---|---|---|---|---|
| DINOv3-ViT-L | 1024 | 21.83 | 24.49 | DINOv2-ViT-S | 384 | 20.30 | 24.64 |
| DINOv3-ViT-B | 768 | 21.83 | 24.35 | DINOv2-ViT-B | 768 | 21.80 | 25.06 |
| DINOv3-ViT-S | 384 | 21.56 | 24.48 | DINOv2-ViT-L | 1024 | **21.84** | **25.77** |
| ResNet-50 | 256 | 21.84 | 25.50 | DINOv2-ViT-G | 1536 | 21.84 | 25.75 |

## A.5 ABLATION ON GNN ARCHITECTURES

**Setup.** We evaluate several standard GNN backbones on topology reasoning. Unless otherwise noted, we use AdamW with cosine LR, DinoV2 ViT-L/14 (1024-d) features, hidden dim $= 64$, dropout $= 0$, ReLU activation, and 100 training epochs. Scores are reported at the best epoch for lane–lane ($\text{TOP}_{ll}$) and lane–TE ($\text{TOP}_{lt}$). The results are shown in Table 5.

**Why further looking into GAT.** Among the tested backbones, GAT consistently showed stronger $\text{TOP}_{ll}$ (lane–lane connectivity), which is empirically the harder subtask. We therefore performed a focused hyperparameter search on GAT, shown in Figure 6. These results are intended to show relative trends: real node perturbations, loss weights ($w_{ll}, w_{lt}$), and other knobs remain untuned.

(1) GAT scales well in head count, with 3-layer, 32-head at hidden=64 achieving the best $\text{TOP}_{lt}$ (24.70). (2) Larger hidden dims (256, 512) do not necessarily help. (3) Decremental widths plateau around $\sim 22.0$. These results confirm GAT's relative advantage on the harder $\text{TOP}_{ll}$ task, though absolute numbers remain improvable with further tuning.

Table 5: Baseline GNN architectures (best epoch). Default hidden dim=64, dropout=0.

| Model (config) | $\text{TOP}_{ll}$ | $\text{TOP}_{lt}$ |
|---|---|---|
| GraphConv (2L) | 21.85 | 19.95 |
| GraphConv (3L) | 10.95 | **23.79** |
| GraphConv (4L) | 21.85 | 22.02 |
| GAT (2L, 4 heads) | 16.22 | 22.61 |
| GAT (2L, 8 heads) | 21.77 | 20.31 |
| GAT (2L, 16 heads) | 18.94 | 22.16 |
| GAT (2L, 16 heads, 100ep) | **21.94** | 21.91 |
| GIN ($\epsilon = 1$, 2L) | 17.80 | 17.14 |
| GIN ($\epsilon = 0.1$, 2L) | 14.79 | 22.02 |
| GIN ($\epsilon = 0.01$, 2L) | 12.22 | 15.11 |
| GIN ($\epsilon = 0.001$, 2L) | 12.65 | 19.76 |
| GraphSAGE (2L) | 21.85 | 12.15 |
| GraphSAGE (3L) | 21.85 | 6.36 |
| GraphSAGE (4L) | 21.87 | 21.13 |
| Transformer (4 heads) | 18.00 | 15.98 |
| Transformer (8 heads) | 16.37 | 20.03 |
| Transformer (16 heads) | 17.59 | 18.94 |

Table 6: The hyper-parameter sutdies on GAT (best epoch). Each cell: *(dropout, heads)* $\rightarrow$ $TOP_{ll}/TOP_{lt}$. DinoV2: ViT-L/14 (1024-d). Hidden dims are per-layer sizes. We highlight the best $TOP_{lt}$ per block.

| Hidden dim | 2 layers | 3 layers | Notes |
|---|---|---|---|
| 64 | $(0.2, 16) \rightarrow 14.54/$**23.60** | $(0.2, 32) \rightarrow 15.81/$**24.70** | Best overall $TOP_{lt}$ at 3L, 32h |
| 128 | $(0.3, 32) \rightarrow 14.27/$**24.47** | $(0.2, 8) \rightarrow 14.94/$**24.48** | Stable $\sim$24.5 |
| 256 | $(0.1, 32) \rightarrow 12.83/$**23.62** | $(0.1, 8) \rightarrow 13.65/21.98$ | 2L > 3L |
| 512 | $(0.1, 32) \rightarrow 14.03/$**22.58** | – | Larger dim no gain |
| $[512, 256]$ | $(0.1, 16) \rightarrow 20.69/$**22.02** | – | Decremental width $\sim$22.0 |
| $[512, 256, 128]$ | – | $(0.1, 32) \rightarrow 10.99/$**22.02** | Deeper decremental $\sim$22.0 |
| $[512, 256, 128, 64]$ | – | – | 4L decremental $\sim$22.0 |

### A.6 SENSITIVITY STUDY ON FUSION WEIGHT ACROSS FEATURE EXTRACTORS AND BASELINE MODELS

We conduct an extended studies over the lane–lane ($w_{ll}$) and lane–TE ($w_{lt}$) fusion weights (Equation 6) to evaluate the robustness of our refinement framework. The search space is defined as

$$w_{ll}, w_{lt} \in \{0.1, 0.2, \ldots, 0.9\}.$$

**DinoV2 backbones.** Table 7 reports the best five results for each DinoV2 feature extractor when combined with TopoNet. We find that larger backbones (ViT-G, ViT-L) consistently achieve stronger and more stable performance than smaller ones (ViT-B, ViT-S). In particular, ViT-L achieves $TOP_{lt} \approx 25.4$ with $TOP_{ll} \approx 22.0$, and ViT-G reaches $TOP_{lt} = 25.75$ at $(w_{ll}, w_{lt}) = (0.6, 0.4)$. By contrast, ViT-B performs best when $w_{ll}$ is small (0.1–0.2) and $w_{lt}$ dominates, while ViT-S exhibits greater variance. Overall, when the backbone and baseline topology reasoning model are fixed, the weights are relatively stable, with a good range observed at $w_{ll} \in \{0.8, 0.9\}$ and $w_{lt} \in \{0.4, 0.5, 0.6\}$.

Table 7: Best five $(w_{ll}, w_{lt})$ results for each DinoV2 model on GATv2 (20250804_193420).

| DinoV2 Model | $w_{ll}$ | $w_{lt}$ | $TOP_{ll} \uparrow$ | $TOP_{lt} \uparrow$ |
|---|---|---|---|---|
| ViT-G | 0.6 | 0.4 | 21.84 | 25.75 |
| | 0.5 | 0.5 | 21.84 | 25.37 |
| | 0.4 | 0.6 | 21.84 | 24.91 |
| | 0.6 | 0.4 | 20.96 | 25.75 |
| | 0.3 | 0.7 | 21.84 | 24.53 |
| ViT-L | 0.6 | 0.4 | 21.96 | 25.39 |
| | 0.6 | 0.4 | 21.84 | 25.39 |
| | 0.5 | 0.5 | 21.96 | 25.14 |
| | 0.7 | 0.3 | 21.96 | 25.04 |
| | 0.5 | 0.5 | 21.84 | 25.14 |
| ViT-B | 0.1 | 0.9 | 21.93 | 23.52 |
| | 0.1 | 0.9 | 21.79 | 23.52 |
| | 0.1 | 0.9 | 21.76 | 23.52 |
| | 0.1 | 0.9 | 21.69 | 23.52 |
| | 0.2 | 0.8 | 21.93 | 23.20 |
| ViT-S | 0.6 | 0.4 | 20.30 | 24.64 |
| | 0.5 | 0.5 | 20.30 | 24.37 |
| | 0.4 | 0.6 | 20.30 | 24.07 |
| | 0.6 | 0.4 | 19.75 | 24.54 |
| | 0.2 | 0.8 | 20.30 | 23.89 |

To further examine whether our refinement module depends on the strength of the DINOv2 backbone, we evaluate TopoRefine using four different DINOv2 variants (ViT-G, ViT-L, ViT-B, and

ViT-S) applied to the SMART topology predictor. These variants span a wide range of capacities and embedding dimensions, from the largest ViT-G to the lightweight ViT-S model. Importantly, only the embedding backbone changes; the base topology predictor and the refinement GNN remain fixed so that we isolate the effect of backbone strength on refinement quality.

As shown in Table 8, all variants achieve similar refinement performance, with $\text{TOP}_{ll}$ consistently around 40 and only modest variation in $\text{TOP}_{lt}$. Even the smallest ViT-S model provides strong improvements, demonstrating that TopoRefine does not rely on large or powerful visual backbones. Instead, the refinement gains come from the structural learning of the SSL-GNN rather than from the capacity of the upstream feature extractor. This further supports our claim that TopoRefine is model-agnostic, lightweight, and robust to different backbone choices.

Table 8: Performance of different DINOv2 embedding variants applied to SMART. All models are evaluated using their tuned best SSL weights.

| DINOv2 Variant | w_ll | w_lt | $\text{TOP}_{ll}$ | $\text{TOP}_{lt}$ |
|---|---|---|---|---|
| ViT-G | 0.8 | 0.6 | 40.02 | 35.51 |
| ViT-L | 0.8 | 0.6 | 40.08 | 35.40 |
| ViT-B | 0.8 | 0.5 | 40.07 | 35.40 |
| ViT-S | 0.8 | 0.1 | 40.00 | 32.85 |

**Other feature extractors.** We further evaluate DinoV3 and ResNet backbones with TopoNet (Table 9). Results indicate similar optimal weight ranges, with ResNet50 reaching $\text{TOP}_{lt} = 25.50$ at $(w_{ll}, w_{lt}) = (0.7, 0.6)$.

Table 9: Tuned results of different feature extractors (with TopoNet GATv2).

| Feature Extractor | $w_{ll}$ | $w_{lt}$ | $\text{TOP}_{ll}$ ↑ | $\text{TOP}_{lt}$ ↑ |
|---|---|---|---|---|
| DinoV3_ViTl | 0.9 | 0.4 | 21.83 | 24.49 |
| DinoV3_ViTb | 0.9 | 0.4 | 21.83 | 24.35 |
| DinoV3_ViTs | 0.9 | 0.4 | 21.56 | 24.48 |
| ResNet50 | 0.7 | 0.6 | 21.84 | 25.50 |

**Different baseline models.** Finally, we tune weights for a broader set of topology reasoning models (Table 10). Across TopoNet, TopoMLP, SMART-OL, Topo2D, and TopoLogic, we observe consistent gains after applying our refinement. While absolute values vary, the effective range of $(w_{ll}, w_{lt})$ remains stable around $(0.8\text{–}0.9, 0.4\text{–}0.6)$.

Table 10: $w_{ll}$ and $w_{lt}$ results of different baseline topology reasoning models after applying TopoRefine.

| Baseline Model | $w_{ll}$ | $w_{lt}$ | $\text{TOP}_{ll}$ ↑ | $\text{TOP}_{lt}$ ↑ |
|---|---|---|---|---|
| TopoNet | 0.9 | 0.6 | 21.84 | 25.77 |
| TopoMLP | 0.8 | 0.6 | 24.35 | 28.68 |
| SMART-OL | 0.8 | 0.6 | 40.08 | 35.40 |
| Topo2D | 0.8 | 0.6 | 23.08 | 27.35 |
| TopoLogic | 0.8 | 0.6 | 23.79 | 27.23 |

**Summary.** Taken together, these results demonstrate that our refinement strategy is broadly applicable. We evaluated it across diverse feature backbones (DinoV2, DinoV3, ResNet) and multiple topology reasoning models (TopoNet, SMART, TopoMLP, SMART-OL, Topo2D, TopoLogic). In nearly all cases, applying our SSL-based refinement with appropriately weights $(w_{ll}, w_{lt})$ yields substantial improvements over the baselines and approaches the theoretical upper bound. Empirically, the best performance is typically achieved with $w_{ll} \in \{0.8, 0.9\}$ and $w_{lt} \in \{0.4, 0.5, 0.6\}$.

### A.7 HOW THE TOP METRIC IS CALCULATED

**Notation.** Let $G = (V, E)$ be the ground-truth graph and $\hat{G} = (\hat{V}, \hat{E})$ the predicted graph. We distinguish lane nodes $V_l$ and traffic-element (TE) nodes $V_t$, with $\hat{V}_l$ and $\hat{V}_t$ their predictions. Distances are measured by Fréchet distance $d_\ell$ for lanes and IoU-based distance $d_t$ for TEs. We evaluate over threshold sets $\mathcal{D}_\ell = \{1, 2, 3\}$ for lanes and $\mathcal{D}_t = \{0.75\}$ for TEs, with a fixed edge-confidence cutoff $c_0 = 0.5$. Node detection confidence is denoted $s(\cdot)$, and edge confidence $r_{uv}$.

**Step 1. Node matching.** For a threshold $D$, predictions are sorted by confidence $s(\cdot)$ and greedily matched to the nearest ground-truth item within distance $D$. Each ground-truth node can be used at most once. We denote the matched sets as $V_l^\star(D)$ and $V_t^\star(D)$ (and their prediction counterparts). All subsequent topology scoring is restricted to these matched sets.

**Step 2. Edge ranking per node.** For a matched vertex $v$, we collect its predicted incident edges with confidence above cutoff $c_0$, i.e. $r_{v\to u} > c_0$. Neighbors are then ranked from high to low by $r_{v\to u}$. We define a binary indicator $y_i(v) = 1$ if the $i$-th predicted neighbor is a true ground-truth neighbor of $v$, and $y_i(v) = 0$ otherwise. From this ranked list we compute the standard average precision (AP):

$$\text{AP}(v) = \frac{1}{|\mathcal{N}(v)|} \sum_i \text{Precision}_v(i)\, y_i(v),$$

where $\mathcal{N}(v)$ is the ground-truth neighbor set of $v$.

**Step 3. Aggregating AP into TOP scores.**

- **Lane–lane (TOP$_{ll}$):** For each lane threshold $D \in \mathcal{D}_\ell$, we compute AP for matched lanes in both directions (row and column of the adjacency). The final score is the average over all lanes and thresholds.

- **Lane–TE (TOP$_{lt}$):** For each pair $(D_\ell, D_t)$, we compute AP for both matched lanes and TEs, and then average over all thresholds.

**Interpretation.**

- Confidence enters twice: once in node matching (higher $s(\cdot)$ matched first), and once in edge ranking ($r_{uv}$).

- Only edges with $r_{uv} > c_0$ are considered in AP computation.

- If predictions are perfect (all nodes matched and all edges correct), then every $\text{AP}(v) = 1$, so both TOP$_{ll}$ and TOP$_{lt}$ equal 1.

## B EXPERIMENTS DURING REBUTTAL PERIOD

### B.1 ABSOLUTE DIFFERENCE OF TOP METRICS IMPROVEMENTS

To facilitate clearer quantitative interpretation, we modify Tables 1 and 2 by replacing percentage-based relative improvements with the corresponding absolute differences between the values before and after adding TopoRefine, as shown in Tables 11 and 12. This change allows direct comparison of metric shifts without dependence on the underlying baseline magnitude.

### B.2 SENSITIVITY ANALYSIS

#### B.2.1 GAUSSIAN PERTURBATION SENSITIVITY ANALYSIS

To examine how sensitive our refinement module is to the perturbation magnitude used in the augmentation step, we conduct a controlled study by varying the Gaussian standard deviation $\sigma$ in the perturbation formulation of Eq. 1. Specifically, we separately sweep the perturbation applied to (1) lane polylines (lane_std) and (2) traffic-element bounding boxes (te_std), and report the effect on both continuous topology metrics (TOP$_{ll}$, TOP$_{lt}$) and discrete connectivity metrics (TJS$_{ll}$, TJS$_{te}$).

Table 11: Comparison of methods on the OpenLane-V2 Subset A dataset using OpenLane-V2 metrics. Best results are shown in bold and the second-best are underlined. Values in parentheses indicate absolute improvements over the corresponding baseline before adding TopoRefine.

| Input type | Method | Venue | TOP$_{ll}$ ↑ | TOP$_{lt}$ ↑ |
|---|---|---|---|---|
| Perspective images | STSU (Can et al., 2022) | ICCV 2021 | 2.9 | 19.8 |
| | VectorMapNet (Liu et al., 2023) | ICML 2023 | 2.7 | 9.2 |
| | MapTR (Liao et al., 2022) | ICLR 2023 | 5.9 | 15.1 |
| | TopoNet (Li et al., 2023) | Arxiv 2023 | 10.9 | 23.8 |
| | TopoMLP (Wu et al., 2024) | ICLR 2024 | 21.6 | 26.9 |
| | Topo2D (Li et al., 2024a) | Arxiv 2024 | 22.3 | 26.2 |
| | RoadPainter (Ma et al., 2024) | ECCV 2024 | 22.8 | 27.2 |
| | TopoFormer (Lv et al., 2025) | CVPR 2025 | 24.1 | 29.5 |
| | TopoPoint (Fu et al., 2025a) | Arxiv 2025 | 28.7 | 30.0 |
| Perspective images + SD maps | TopoOSMR (Zhang et al., 2024) | IROS 2024 | 17.1 | 26.8 |
| | SMERF (Luo et al., 2024) | ICRA 2024 | 15.4 | 25.4 |
| | TopoLogic (Fu et al., 2024) | NeurIPS 2024 | 23.9 | 25.4 |
| | RoadPainter (Ma et al., 2024) | ECCV 2024 | 29.6 | 29.5 |
| Perspective images + Map priors | SMART (TopoNet) (Ye et al., 2025) | ICRA 2025 | 27.5 | 33.1 |
| | SMART (TopoMLP) (Ye et al., 2025) | ICRA 2025 | 37.0 | 33.0 |
| Perspective images | TopoNet + TopoRefine | | 21.8 (+10.9) | 25.8 (+2.0) |
| | TopoMLP + TopoRefine | | 24.4 (+2.8) | 28.7 (+1.8) |
| | Topo2D + TopoRefine | Ours | 24.3 (+2.0) | 27.3 (+1.1) |
| | TopoLogic + TopoRefine | | 24.5 (+0.6) | 27.2 (+1.8) |
| | SMART (TopoMLP) + TopoRefine | | **40.1** (+3.1) | **35.4** (+2.4) |

Table 12: Combined evaluation across baselines. We report the detection-conditioned upper bound (UB), the gap to this bound (margin; smaller is better), and discrete graph quality measured by TJS (larger is better). "After" results now include absolute changes relative to "Before" values.

| Method | UB (TOP) | | margin$_{ll}$ ↓ | | margin$_{lt}$ ↓ | | TJS$_{ll}$ (%) ↑ | | TJS$_{lt}$ (%) ↑ | |
|---|---|---|---|---|---|---|---|---|---|---|
| | TOP$_{ll}$ | TOP$_{lt}$ | Bef. | Aft. | Bef. | Aft. | Bef. | Aft. | Bef. | Aft. |
| TopoNet | 24.4 | 28.8 | 13.4 | $2.5_{-10.9}$ | 7.3 | $3.0_{-4.3}$ | 16.3 | $32.9_{+16.6}$ | 31.0 | $55.8_{+24.8}$ |
| TopoMLP | 25.0 | 31.4 | 3.3 | $0.7_{-2.6}$ | 4.4 | $2.7_{-1.7}$ | 18.5 | $59.7_{+41.2}$ | 31.5 | $59.8_{+28.3}$ |
| Topo2D | 25.0 | 29.8 | 2.8 | $0.7_{-2.1}$ | 3.6 | $2.5_{-1.1}$ | 18.9 | $55.0_{+36.1}$ | 34.5 | $51.5_{+17.0}$ |
| TopoLogic | 26.0 | 30.6 | 2.0 | $1.5_{-0.5}$ | 5.3 | $3.3_{-2.0}$ | 21.0 | $43.9_{+22.9}$ | 32.5 | $53.3_{+20.8}$ |
| SMART (TopoMLP) | 40.9 | 38.9 | 3.9 | $0.8_{-3.1}$ | 5.9 | $3.5_{-2.4}$ | 38.1 | $87.7_{+49.6}$ | 36.6 | $82.8_{+46.2}$ |

For each configuration, we retrain only the refinement module while keeping the underlying topology predictor fixed, thereby isolating the effect of Gaussian noise strength alone. The perturbation scales span a wide range (lane_std $\in \{0.1, 0.3, 0.5, 0.7, 0.9\}$ and te_std $\in \{4, 6, 8, 10, 12\}$), covering both under-perturbed and over-perturbed regimes. The results are summarized in Table 13.

Table 13: Sensitivity analysis of Gaussian perturbation strength. Performance remains stable across a wide range of noise scales, indicating that the refinement module is robust to the choice of $\sigma$.

| Lane std | TE std | $\text{TOP}_{ll}$ | $\text{TOP}_{lt}$ | $\text{TJS}_{ll}$ | $\text{TJS}_{te}$ |
|---|---|---|---|---|---|
| 0.9 | 12 | 21.84 | 25.50 | 32.92 | 51.68 |
| 0.7 | 12 | 21.84 | 25.71 | 32.92 | 53.29 |
| 0.5 | 12 | 21.84 | 25.76 | 32.92 | 56.32 |
| 0.3 | 12 | 21.84 | 25.79 | 32.92 | 55.81 |
| 0.1 | 12 | 21.84 | 25.74 | 32.92 | 53.49 |
| 0.1 | 10 | 21.83 | 25.67 | 32.86 | 54.47 |
| 0.1 | 8 | 21.84 | 25.71 | 32.92 | 53.13 |
| 0.1 | 6 | 21.83 | 25.63 | 32.92 | 52.67 |
| 0.1 | 4 | 21.84 | 25.78 | 32.92 | 55.77 |

Across all perturbation strengths, the continuous metrics $\text{TOP}_{ll}$ and $\text{TOP}_{lt}$ remain effectively unchanged (variations $< 0.3$), demonstrating that the refinement module does not depend on a specific Gaussian scale. The discrete connectivity metrics ($\text{TJS}_{ll}$, $\text{TJS}_{te}$) also remain highly stable, fluctuating within a narrow band despite nearly an order-of-magnitude change in perturbation strength. This confirms that the perturbation serves as a generic and smooth augmentation mechanism rather than a model of real-world prediction errors.

### B.2.2 DETECTION AND TOPOLOGY CONFIDENCE THRESHOLD

Both TJS and TOP metrics change systematically with the detection and topology thresholds, since thresholding directly determines the sparsity of the predicted graph used in evaluation. Lower thresholds retain more candidate edges, increasing recall and typically improving both TOP and TJS, whereas higher thresholds prune edges more aggressively and therefore reduce these metrics. This trend is clearly visible in Table 14 and aligns with the definition of both metrics as overlap-based measures of connectivity quality.

Crucially, however, the *relative behavior* of the metric remains stable across a wide threshold range: configurations that perform well under the default settings continue to do so even as thresholds vary. These results were obtained using our standard setup—DINOv2-ViT-L embeddings with TopoNet as the base topology predictor—ensuring that all comparisons are made under a consistent evaluation pipeline. Thus, while absolute scores naturally shift with thresholding (as expected for overlap-based metrics), the comparative ranking of configurations remains unchanged. This demonstrates that the evaluation is robust and that our conclusions are not sensitive to the particular choice of threshold values.

### B.3 RESULTS ON SUBSET B

We only report TopoNet results on Subset B because it is the only topology reasoning model that publicly provides pretrained checkpoints for this subset, enabling a fair and consistent evaluation of our post-hoc refinement module.

These results demonstrate that TopoRefine brings substantial improvements on Subset B despite leaving the underlying TopoNet model completely unchanged. This aligns with the results on the Subset A. The refined predictions closely approach the detection-conditioned upper bound across all metrics, indicating that improving discrete connectivity does not require retraining or modifying the base topology model. This reinforces the generality of our post-hoc design and shows that the refinement module transfers effectively across dataset subsets.

Table 14: Sensitivity analysis of detection and topology confidence thresholds for computing TOP and TJS. Results are generated using the default setup DINOv2-ViTL embeddings with TopoNet as the base topology predictor, ensuring all comparisons follow a consistent and well-performed evaluation pipeline.

| Det. Th. | Top. Th. | $\text{TOP}_{ll}$ | $\text{TOP}_{lt}$ | $\text{TJS}_{ll}$ | $\text{TJS}_{te}$ |
|---|---|---|---|---|---|
| 0.1 | 0.1 | 33.5 | 25.3 | 42.1 | 60.5 |
| 0.1 | 0.3 | 35.0 | 26.1 | 39.4 | 68.3 |
| 0.1 | 0.5 | 33.1 | 24.1 | 38.1 | 36.6 |
| 0.1 | 0.7 | 27.3 | 22.6 | 34.1 | 21.7 |
| 0.1 | 0.9 | 14.2 | 21.8 | 21.1 | 14.0 |
| 0.3 | 0.1 | 31.1 | 19.9 | 42.1 | 58.6 |
| 0.3 | 0.3 | 32.3 | 20.5 | 39.4 | 68.3 |
| 0.3 | 0.5 | 30.7 | 18.6 | 38.1 | 36.6 |
| 0.3 | 0.7 | 25.5 | 17.1 | 34.1 | 21.7 |
| 0.3 | 0.9 | 13.3 | 16.3 | 21.1 | 14.0 |
| 0.5 | 0.1 | 27.8 | 16.4 | 45.8 | 22.7 |
| 0.5 | 0.3 | 28.7 | 16.9 | 41.1 | 28.3 |
| 0.5 | 0.5 | 27.3 | 15.1 | 39.0 | 36.7 |
| 0.5 | 0.7 | 23.0 | 13.6 | 34.3 | 21.5 |
| 0.5 | 0.9 | 12.0 | 12.9 | 20.8 | 13.8 |
| 0.7 | 0.1 | 23.4 | 12.7 | 42.3 | 29.0 |
| 0.7 | 0.3 | 24.0 | 13.0 | 37.8 | 29.5 |
| 0.7 | 0.5 | 22.9 | 11.4 | 35.8 | 25.9 |
| 0.7 | 0.7 | 19.6 | 10.0 | 31.6 | 21.1 |
| 0.7 | 0.9 | 10.4 | 9.3 | 19.7 | 13.5 |
| 0.9 | 0.1 | 15.4 | 7.7 | 32.2 | 8.8 |
| 0.9 | 0.3 | 16.6 | 7.9 | 29.2 | 7.8 |
| 0.9 | 0.5 | 16.0 | 6.7 | 27.9 | 6.8 |
| 0.9 | 0.7 | 14.0 | 6.6 | 25.0 | 6.8 |
| 0.9 | 0.9 | 8.0 | 5.0 | 16.6 | 5.3 |

Table 15: Results on OpenLane-V2 Subset B. TopoRefine consistently improves both continuous ($\text{TOP}_{ll}$, $\text{TOP}_{lt}$) and discrete ($\text{TJS}_{ll}$, $\text{TJS}_{te}$) topology metrics over the original TopoNet baseline, and achieves performance close to the detection-conditioned upper bound.

| Method | $\text{TOP}_{ll}$ | $\text{TOP}_{lt}$ | $\text{TJS}_{ll}$ | $\text{TJS}_{te}$ |
|---|---|---|---|---|
| TopoNet (w/o TopoRefine) | 6.7 | 16.7 | 9.0 | 35.6 |
| TopoNet (w/ TopoRefine) | 19.5 | 17.8 | 42.3 | 56.8 |
| TopoNet Upper Bound | 21.2 | 20.3 | 48.5 | 66.7 |

### B.4 INFERENCE LATENCY AND MEMORY BREAKDOWN

For a detailed breakdown of runtime and memory cost, we benchmark the full TopoRefine inference pipeline on a representative OpenLane-V2 Subset B with TopoNet as an example. We run everything on a single H200 GPU. The graph contains 200 lane nodes, 100 traffic-element (TE) nodes, 39,800 candidate lane–lane edges, and 20,000 candidate lane–TE edges. This setting reflects the typical scale of real-world scenes and stress-tests the refinement module under dense connectivity.

Table 16 reports the latency of each component. The total inference cost of TopoRefine is **6.43 ms**, with the majority of time spent in GNN message passing (**80.9%**). Feature extraction/refinement and the fusion (edge prediction) stage contribute only a small fraction of the overall runtime.

Table 17 summarizes the corresponding memory overhead. The refinement module incurs only **0.3 MB** of additional memory usage, with no increase during GNN message passing and minimal

Table 16: Latency breakdown of TopoRefine on a representative OpenLane-V2 scene (200 lanes, 100 TEs).

| Component | Latency (ms) | Percentage |
|---|---|---|
| Feature extraction / refinement | 0.21 | 3.2% |
| GNN message passing | 5.21 | 80.9% |
| Fusion (edge prediction) | 0.64 | 9.9% |
| **Total** | **6.43** | **100%** |

additional cost for feature extraction and edge fusion. This confirms that TopoRefine is lightweight and suitable for deployment in real-time systems.

Table 17: Memory overhead of each inference component.

| Component | Memory Overhead |
|---|---|
| Feature extraction / refinement | +0.1 MB |
| GNN message passing | +0.0 MB |
| Fusion (edge prediction) | +0.2 MB |
| **Total** | **+0.3 MB** |

Overall, these measurements demonstrate that TopoRefine introduces only a small and predictable overhead. The method remains fast, memory-efficient, and scales smoothly with graph size, making it practical as a post-hoc refinement module for modern topology reasoning pipelines.

## B.5    PERFORMANCE OF USING WEAKER EMBEDDING BACKBONES (RESNET-50)

To verify that TopoRefine does not rely on strong visual backbones and remains effective even under substantially weaker feature representations, we replace the DINOv2-ViT-L embedding used in the main experiments with a significantly weaker ResNet-50 encoder and apply the same refinement pipeline to all topology reasoning models. The results are summarized in Table 18.

Across all baselines, TopoRefine continues to yield consistent improvements over the original (non-refined) models, even when feature quality is markedly degraded. Although DINOv2 features produce stronger absolute scores—as expected—TopoRefine still provides large relative gains under ResNet-50. This confirms that the refinement effectiveness is not tied to a particular backbone, and that TopoRefine does not depend on using DINOv2 to indirectly boost detection or topology accuracy. Instead, it operates purely as a post-hoc, model-agnostic GNN refinement independent of the upstream feature extractor.

Table 18: Comparison of refinement performance using ResNet-50 versus DINOv2-ViT-L embeddings, alongside the original (non-refined) outputs. Even with weaker ResNet embeddings, TopoRefine consistently improves both continuous (TOP$_{ll}$, TOP$_{lt}$) and discrete (TJS$_{ll}$, TJS$_{te}$) metrics, demonstrating that its effectiveness does not depend on strong backbone features.

| Model | TOP$_{ll}$ | TOP$_{lt}$ | TJS$_{ll}$ | TJS$_{te}$ |
|---|---|---|---|---|
| **ResNet-50 Embedding** | | | | |
| TopoNet | 21.8 | 25.5 | 32.9 | 51.4 |
| TopoMLP | 23.1 | 28.3 | 59.7 | 57.8 |
| Topo2D | 23.1 | 26.8 | 55.0 | 46.2 |
| TopoLogic | 23.8 | 26.8 | 43.9 | 50.1 |
| SMART | 36.6 | 35.1 | 87.7 | 81.4 |
| **DINOv2-ViT-L Embedding** | | | | |
| TopoNet | 21.8 | 25.8 | 32.9 | 55.8 |
| TopoMLP | 24.4 | 28.7 | 59.7 | 59.8 |
| Topo2D | 24.3 | 27.3 | 55.0 | 51.5 |
| TopoLogic | 24.5 | 27.2 | 43.9 | 53.3 |
| SMART | 40.1 | 35.4 | 87.7 | 82.8 |
| **Original Model (No Refinement)** | | | | |
| TopoNet | 10.9 | 23.8 | 16.3 | 31.0 |
| TopoMLP | 21.6 | 26.9 | 18.5 | 31.5 |
| Topo2D | 22.3 | 26.2 | 18.9 | 34.5 |
| TopoLogic | 23.9 | 25.4 | 21.0 | 32.5 |
| SMART | 37.0 | 33.0 | 38.1 | 36.6 |

