# OpenReview forum: "Rethinking Driving Topology Reasoning: Plug-and-Play Discrete Graph Refinement"
_ICLR.cc/2026/Conference — Submitted to ICLR 2026_

### Official Review · Reviewer_VE9m · 2025-10-24

**Soundness:** 2
**Presentation:** 2
**Contribution:** 2
**Rating:** 2
**Confidence:** 5

**Summary:**

This paper proposes TopoRefine, a plug-and-play refinement module designed to improve the discrete graph quality in driving topology reasoning. The method employs a self-supervised GNN to recalibrate edge confidences predicted by existing topology models and introduces a new evaluation metric, the Topology Jaccard Score (TJS), to measure discrete connectivity. The authors claim that TopoRefine is model-agnostic and can enhance both continuous and discrete topology metrics without retraining. Experiments on the OpenLane-V2 dataset show consistent improvements across multiple baselines.

**Strengths:**

1. The proposed refinement module is model-agnostic and can be easily applied to existing methods.
2. The experiments show consistent quantitative improvements across various baselines.

**Weaknesses:**

1. The definitions and conceptual differences between continuous graphs and discrete graphs are insufficiently clear, leading to a vague motivation for why this distinction is crucial.

2. The proposed Topology Jaccard Score (TJS) is essentially equivalent to a standard Graph IoU metric and therefore does not provide substantial new conceptual insight. Furthermore, IoU-based measures are inherently sensitive to the threshold used to determine true and false positives; however, the paper lacks any sensitivity analysis or discussion regarding this critical aspect.

3. As described in Sections 3 and 5.2, the framework depends on centerline polylines and traffic element features extracted from strong pretrained backbones such as DINOv2-ViT-L. Consequently, the method functions largely as a post-hoc “plug-and-play” refinement on top of existing lane detection pipelines, introducing additional complexity. Moreover, the integration between this refinement module and existing lane detection or topology reasoning models appears limited, without demonstrating meaningful synergistic effects.

4. Line 301: The paper claims that OpenLane-V2 uses a specific matching formula based on the Hungarian Matching Algorithm, but the corresponding equation seems incorrect and inconsistent with the original OpenLane-V2 implementation.
5. In Table 1, the TOP_ll metric has undergone an update between benchmark versions, yet the paper does not clearly indicate or distinguish which version was used. This lack of clarification casts doubt on the reported performance gains.

6. The evaluation is restricted to OpenLane-V2 Subset A only, with no validation on Subset B.

7. The perturbation strategy in Table 3 yields only marginal performance improvements, suggesting that its contribution to the overall framework may be minimal.

**Questions:**

N/A

---

> ### Author Response · Authors · 2025-11-24
> **Reviewer VE9m -  Response [1/7]**
>
> Comment:
> > The definitions and conceptual differences between continuous graphs and discrete graphs are insufficiently clear, leading to a vague motivation for why this distinction is crucial.
>
> We thank the reviewer for pointing this out. We agree that the distinction between continuous and discrete graphs is central to the motivation of our work and that the original writing could be clearer. In the revision, we have updated the problem statement to explicitly define these two graph representations and to clarify why discrete connectivity—rather than continuous confidence scores—is what downstream planning ultimately depends on. **The revised explanation is now included in Page 1 line 49 to Page 2 line 75.**
>
> > For clarity, we distinguish continuous from discrete topology graphs. Continuous graphs assign confidence scores to all candidate connections, producing a dense structure optimized by existing models. Discrete graphs, however, are the sparse binary connectivities required by downstream planning. Current methods convert continuous scores to discrete edges using a fixed threshold, but improvements in the continuous domain do not necessarily yield better discrete graphs. This gap motivates our focus on explicitly evaluating and refining discrete topology.

---

> ### Author Response · Authors · 2025-11-24
> **Reviewer VE9m - Response [2/7]**
>
> Comment:
> > The proposed Topology Jaccard Score (TJS) is essentially equivalent to a standard Graph IoU metric and therefore does not provide substantial new conceptual insight. Furthermore, IoU-based measures are inherently sensitive to the threshold used to determine true and false positives; however, the paper lacks any sensitivity analysis or discussion regarding this critical aspect.
>
> We thank the reviewer for the thoughtful comment. While TJS is inspired by Jaccard-style overlap measures, it is not equivalent to a “Graph IoU’’ metric. (To the best of our knowledge, no established “Graph IoU’’ formulation resolves the node-misalignment issue in autonomous-driving graphs, which we believe motivated the reviewer’s question.) Standard IoU assumes a shared node set or a one-to-one node correspondence, but in autonomous driving the predicted and ground-truth nodes lie at different coordinates and must first be aligned.
>
> TJS explicitly incorporates the official OpenLane-V2 detection-aware matching protocol and evaluates overlap only between edges whose endpoints are successfully matched. This matching step is essential and makes TJS fundamentally different from applying IoU directly to edge sets without node alignment.
>
> Regarding threshold sensitivity, we agree this is important. Beyond using the standard 0.5 threshold adopted in prior work, we have added a comprehensive sensitivity analysis varying detection thresholds, topology thresholds, and matching radii. As shown in **Appendix B.2.2 (and in the table in Reviewer 2Uto-Response [1/5])**, the absolute TJS values shift with thresholding—as expected for any overlap-based metric—but the relative ranking of methods remains stable across a wide range of settings.
>
> We have updated Section 5.1 to clarify the distinction between TJS and IoU-style metrics and included the new threshold-sensitivity experiments in the appendix.
>
> ### Sensitivity analysis of detection and topology thresholds
> Results use the default setup (DINOv2-ViT-L embeddings + TopoNet).
>
> | Det. Th. | Top. Th. | TOP_ll | TOP_lt | TJS_ll | TJS_te |
> |---------:|---------:|-------:|-------:|--------:|--------:|
> | 0.1 | 0.1 | 33.5 | 25.3 | 42.1 | 60.5 |
> | 0.1 | 0.3 | 35.0 | 26.1 | 39.4 | 68.3 |
> | 0.1 | 0.5 | 33.1 | 24.1 | 38.1 | 36.6 |
> | 0.1 | 0.7 | 27.3 | 22.6 | 34.1 | 21.7 |
> | 0.1 | 0.9 | 14.2 | 21.8 | 21.1 | 14.0 |
> | 0.3 | 0.1 | 31.1 | 19.9 | 42.1 | 58.6 |
> | 0.3 | 0.3 | 32.3 | 20.5 | 39.4 | 68.3 |
> | 0.3 | 0.5 | 30.7 | 18.6 | 38.1 | 36.6 |
> | 0.3 | 0.7 | 25.5 | 17.1 | 34.1 | 21.7 |
> | 0.3 | 0.9 | 13.3 | 16.3 | 21.1 | 14.0 |
> | 0.5 | 0.1 | 27.8 | 16.4 | 45.8 | 22.7 |
> | 0.5 | 0.3 | 28.7 | 16.9 | 41.1 | 28.3 |
> | 0.5 | 0.5 | 27.3 | 15.1 | 39.0 | 36.7 |
> | 0.5 | 0.7 | 23.0 | 13.6 | 34.3 | 21.5 |
> | 0.5 | 0.9 | 12.0 | 12.9 | 20.8 | 13.8 |
> | 0.7 | 0.1 | 23.4 | 12.7 | 42.3 | 29.0 |
> | 0.7 | 0.3 | 24.0 | 13.0 | 37.8 | 29.5 |
> | 0.7 | 0.5 | 22.9 | 11.4 | 35.8 | 25.9 |
> | 0.7 | 0.7 | 19.6 | 10.0 | 31.6 | 21.1 |
> | 0.7 | 0.9 | 10.4 | 9.3  | 19.7 | 13.5 |
> | 0.9 | 0.1 | 15.4 | 7.7 | 32.2 | 8.8 |
> | 0.9 | 0.3 | 16.6 | 7.9 | 29.2 | 7.8 |
> | 0.9 | 0.5 | 16.0 | 6.7 | 27.9 | 6.8 |
> | 0.9 | 0.7 | 14.0 | 6.6 | 25.0 | 6.8 |
> | 0.9 | 0.9 | 8.0  | 5.0 | 16.6 | 5.3 |

---

> ### Author Response · Authors · 2025-11-24
> **Reviewer VE9m - Response [3/7]**
>
> Comments:
> > As described in Sections 3 and 5.2, the framework depends on centerline polylines and traffic element features extracted from strong pretrained backbones such as DINOv2-ViT-L. Consequently, the method functions largely as a post-hoc “plug-and-play” refinement on top of existing lane detection pipelines, introducing additional complexity. Moreover, the integration between this refinement module and existing lane detection or topology reasoning models appears limited, without demonstrating meaningful synergistic effects.
>
> We thank the reviewer for the insightful comments. While our main experiments use DINOv2-ViT-L features, these embeddings are not fed into any pretrained topology reasoning model; they are used only for training the self-supervised GNN in our refinement module. The refinement network itself is a lightweight two-layer GNN applied post-hoc to the predictions of existing models.
>
> To directly address the concern that our gains may rely on strong DINOv2 features, we added an ablation using a much weaker ResNet-50 embedding (Appendix B.5). TopoRefine still delivers substantial improvements across all baselines, demonstrating that the method does not depend on high-capacity backbones and remains effective under significantly reduced feature quality.
>
> Regarding the comment that integration with existing pipelines appears limited: this behavior is intentional. TopoRefine is designed to be a plug-and-play refinement module that **requires no retraining, architectural modification, or loss redesign** for any existing topology reasoning model. The only required inputs are predicted centerline polylines and traffic-element features (e.g., JSON outputs), making integration simple and practical.
>
> On complexity: the refinement stage is small and lightweight, and actually reduces overall pipeline complexity by eliminating the need for each individual model to incorporate discrete-graph supervision into its own training.
>
> The synergy of the method comes from its **model-agnostic** applicability and **consistent gains** across all backbones and architectures, rather than from architectural coupling. We have strengthened the introduction to clarify this motivation, and the **ResNet-50 ablation in Appendix B.5**, also the table below further confirms that TopoRefine’s benefits do not stem from DINOv2 or detector improvements.
>
> ### **ResNet vs. DINOv2 Embedding Ablation**
>
> Comparison of refinement performance using **ResNet-50** versus **DINOv2-ViT-L** embeddings, alongside the original (non-refined) outputs. Even with weaker ResNet features, TopoRefine consistently improves both continuous (TOP_ll, TOP_lt) and discrete (TJS_ll, TJS_te) metrics, demonstrating that its effectiveness does not depend on strong backbone embeddings.
>
> | **Model** | **TOP_ll** | **TOP_lt** | **TJS_ll** | **TJS_te** |
> |-----------|-----------:|-----------:|------------:|------------:|
> | **ResNet-50 Embedding** |||||
> | TopoNet    | 21.8 | 25.5 | 32.9 | 51.4 |
> | TopoMLP    | 23.1 | 28.3 | 59.7 | 57.8 |
> | Topo2D     | 23.1 | 26.8 | 55.0 | 46.2 |
> | TopoLogic  | 23.8 | 26.8 | 43.9 | 50.1 |
> | SMART      | 36.6 | 35.1 | 87.7 | 81.4 |
> | **DINOv2-ViT-L Embedding** |||||
> | TopoNet    | 21.8 | 25.8 | 32.9 | 55.8 |
> | TopoMLP    | 24.4 | 28.7 | 59.7 | 59.8 |
> | Topo2D     | 24.3 | 27.3 | 55.0 | 51.5 |
> | TopoLogic  | 24.5 | 27.2 | 43.9 | 53.3 |
> | SMART      | 40.1 | 35.4 | 87.7 | 82.8 |
> | **Original Model (No Refinement)** |||||
> | TopoNet    | 10.9 | 23.8 | 16.3 | 31.0 |
> | TopoMLP    | 21.6 | 26.9 | 18.5 | 31.5 |
> | Topo2D     | 22.3 | 26.2 | 18.9 | 34.5 |
> | TopoLogic  | 23.9 | 25.4 | 21.0 | 32.5 |
> | SMART      | 37.0 | 33.0 | 38.1 | 36.6 |

---

> ### Author Response · Authors · 2025-11-24
> **Reviewer VE9m - Response [4/7]**
>
> Comment:
> > Line 301: The paper claims that OpenLane-V2 uses a specific matching formula based on the Hungarian Matching Algorithm, but the corresponding equation seems incorrect and inconsistent with the original OpenLane-V2 implementation.
>
> We thank the reviewer for the comment. To clarify, our paper does not claim that OpenLane-V2 provides the Hungarian matching formula.
>
> All our experiments strictly follow the official OpenLane-V2 evaluator and matching implementation without any re-implementation or modification.

---

> ### Author Response · Authors · 2025-11-24
> **Reviewer VE9m - Response [5/7]**
>
> Comment:
> > In Table 1, the TOP_ll metric has undergone an update between benchmark versions, yet the paper does not clearly indicate or distinguish which version was used. This lack of clarification casts doubt on the reported performance gains.
>
> We thank the reviewer for pointing this out. All results in Table 1 are evaluated using the latest OpenLane-V2 benchmark (v2.1.0), and all baselines are taken from papers that also report numbers under this updated version. This is why, for example, TopoNet’s TOP_ll is 10.8 instead of the older 4.1 reported in its original paper.
>
> Since recent works have already adopted OLV2 v2.1.0 as the default, we followed the same convention and did not explicitly restate the version. We will clarify the benchmark version in the revised manuscript (Section 5.1 Evaluation Metrics) for completeness.
>
> Importantly, this clarification does not affect our reported gains, as all methods are compared under the same official v2.1.0 evaluation.

---

> ### Author Response · Authors · 2025-11-24
> **Reviewer VE9m - Response [6/7]**
>
> Comment:
> > The evaluation is restricted to OpenLane-V2 Subset A only, with no validation on Subset B.
>
> We thank the reviewer for raising this point. Following the suggestion, we have added evaluations on OpenLane-V2 Subset B. Since only TopoNet publicly provides pretrained checkpoints for this subset, we report results using this model to ensure a fair and consistent comparison. As shown below (and in **Table 15 of Appendix B.3**), TopoRefine brings substantial improvements over the TopoNet baseline on both continuous (TOP) and discrete (TJS) metrics. The gains closely match those observed on Subset A, confirming that our post-hoc refinement generalizes well across dataset subsets. The manuscript has been updated accordingly.
>
> ### Results on Subset B
>
> | Method                 | TOP_ll | TOP_lt | TJS_ll | TJS_te |
> |------------------------|-------:|-------:|-------:|-------:|
> | TopoNet (w/o Refine)   |   6.7  |  16.7  |   9.0  |  35.6  |
> | TopoNet (w/ Refine)    |  19.5  |  17.8  |  42.3  |  56.8  |
> | TopoNet Upper Bound    |  21.2  |  20.3  |  48.5  |  66.7  |

---

> ### Author Response · Authors · 2025-11-24
> **Reviewer VE9m - Response [7/7]**
>
> Comment:
> > The perturbation strategy in Table 3 yields only marginal performance improvements, suggesting that its contribution to the overall framework may be minimal.
>
> We thank the reviewer for the comment. The perturbation tested in Table 3 refers only to the real-node perturbation, which is a very small component of our overall framework. Our full method includes self-supervised GNN training, fake-node perturbation for negative sample construction, real-node perturbation for augmentation, and the adaptive BCE loss.
>
> The ablation in Table 3 isolates real-node perturbation purely for completeness; hence its marginal effect is expected and does not reflect the contribution of the complete framework. We will revise the description in the manuscript (Section 5.4 Real Node Perturbation) to clarify this point.

---

> > ### Comment · Reviewer_VE9m · 2025-11-25
> >
> > Thanks for your response.  I truly appreciate the significant efforts you have invested in this rebuttal. Nevertheless, certain inherent weaknesses in the argument still persist:
> >
> > 1. From the perspective of final results, Continuous graphs convert the confidence score of each vertex into a 0/1 binary value based on a threshold. Discrete topology graphs also yield such binary values—they use thresholds to obtain these binary values themselves. No strong motivation for this design choice is observed.
> >
> > 2. The term "Graph IoU" may not be sufficiently accurate. Formally, we define a predicted graph connection matrix G with a shape of \(N * N\) and the ground truth (GT) as \(G'\), where each element is binary. I believe the calculation method should be (Intersection(G, G') / Union(G, G')). Furthermore, threshold sensitivity experiments indicate that the final results are highly dependent on the threshold setting.
> >
> > 3. The complexity here refers to the need to train the topology on top of existing lane/traffic detection systems, which introduces training complexity and additional computational overhead. Moreover, the upper performance bound of this approach is constrained by the underlying lane/traffic detection, making it an inelegant solution.
> >
> > In summary, I will keep my original scores.

---

> > > ### Author Response · Authors · 2025-11-30
> > > **R1: Motivation for Continuous Graph Prediction and Thresholding**
> > >
> > > Comment:
> > > > From the perspective of final results, Continuous graphs convert the confidence score of each vertex into a 0/1 binary value based on a threshold. Discrete topology graphs also yield such binary values—they use thresholds to obtain these binary values themselves. No strong motivation for this design choice is observed.
> > >
> > > We thank the reviewers for the thoughtful feedback. We totally understand where the confusion or unclear motivation comes from. It is true that, similar to the base topology reasoning model, our refinement module also outputs a continuous graph and then does the thresholding. Therefore, our motivation seems “not very strong” because we also require thresholding to obtain the discrete graph instead of directly outputting the discrete graph. Directly outputting the discrete graph seems to be the most straightforward approach. However, this approach is theoretically infeasible because predicting the discrete graph directly is mathematically represented by an adjacency matrix with either zeros or ones, and those discrete adjacency matrices would prevent the gradient from being computed and thus prevent the back-propagation. Therefore, in the graph learning community, the common practice is computing the gradient on a continuous relaxed graph predicted by the model, that is, predicting a continuous confidence for each edge and computing the loss function on top of this continuous relaxed graph for back-propagation. Apparently, the existing base topology reasoning model (e.g., TopoNet or TopoMLP) all closely follow this practice, and so does TopoRefine, since predicting a continuous relaxed topology graph is required for back-propagation.
> > >
> > > Given this continuity constraint, the only approach we can take to obtain a better discrete graph is to predict a more accurate confidence score, such that we can obtain a better discrete graph after thresholding. Following this idea, the best we can do is to predict the confidence of GT edges as 1 and predict the confidence of non-GT edges as 0, which is used to measure the upper-bound performance in Table 2 UB(TOP). As you can see from Column 4 and Column 6 of Table 2, after being refined by TopoRefine, the margin to upper-bound performance margin_{ll} and margin_{lt} are all close to zeros for all baseline approaches. In other words, our self-supervised TopoRefine has almost done the best we can do for refining the model predictions of all base topology reasoning models, which correctly predict the confidence of GT edges as close as possible to 1 and predict the confidence of non-GT edges as close as possible to 0. For the same reason, our TJS score after refinement (at Column 8 and Column 10  of Table 2) achieves a big improvement compared to the raw predictions from the base topology reasoning model, which demonstrates the superior quality of our discrete topology graph after thresholding. In a word, predicting a continuous topology graph with more calibrated edge confidence and then doing thresholding is the only possible design choice we can take, as long as back-propagation is needed.

---

> > > ### Author Response · Authors · 2025-11-30
> > > **R2: Clarifying the Distinction Between IoU and the Proposed Topology Jaccard Similarity**
> > >
> > > Comment:
> > > > The term "Graph IoU" may not be sufficiently accurate. Formally, we define a predicted graph connection matrix G with a shape of (N * N) and the ground truth (GT) as (G'), where each element is binary. I believe the calculation method should be (Intersection(G, G') / Union(G, G')). Furthermore, threshold sensitivity experiments indicate that the final results are highly dependent on the threshold setting.
> > >
> > > We sincerely appreciate the reviewer for this insightful comment. We understand that the reviewer may have some concerns about our contribution in terms of proposing this TJS metric because the proposed TJS metric is somewhat similar to IoU. We do agree that TJS is similar to IoU because both TJS and IoU are inspired by Jaccard Similarity, which is why we call it Topology Jaccard Similarity (TJS). Jaccard Similarity is a general metric that compares the similarity between two sets, and IoU is a common metric for image segmentation or object detection in computer vision. However, when we come to evaluating the distance between two discrete graphs (i.e., predicted graph and ground-truth graph), the Jaccard-Similarity-styled metric is not common or has never been used in graph theory. In the graph community, the most popular distance metric between two discrete graphs is Graph Edit Distance (GED). However, as mentioned in Section 4.2, GED does not align with our goal of evaluating the quality of driving topology prediction.
> > >
> > > Since our central need is to solely evaluate the discrete topology connection in the driving scene, the Jaccard-Similarity-styled metric comes to our mind. As the reviewer mentioned, the calculation seems as straightforward as “(Intersection(G, G') / Union(G, G'))”, supposing that G and G’ are the adjacency matrices of the predicted graph and ground-truth graph. However, in the driving topology reasoning task, G and G’ have different shapes. G’ has the shape of NxN and nodes {0, 1, 2, ..., N} where N is the total number of ground-truth detections; G has the shape of MxM and nodes {0, 1, 2,..., M} where M is the total number of detections being proposed. Because all the existing base topology reasoning models adopt DETR-style detection module, M in most cases is greater than N. The question that arises here is that node 0 from G may not correspond to node 0 from G’. Such node matching is required; otherwise, we cannot compute “(Intersection(G, G') / Union(G, G'))”. For this purpose, we carefully and specifically design this node matching algorithm for the driving scene topology task as described in line 319 - line 325 in the manuscript. This node matching algorithm is not trivial and is specifically designed for the driving topology reasoning task, such that the Jaccard Similarity is allowed to be computed as “(Intersection(G, G') / Union(G, G'))”. That’s why we call it Topology Jaccard Similarity. To the best of our knowledge, we are the first work introducing such a Jaccard-Similarity-styled metric to evaluate the quality of topology prediction, together with the node matching algorithm specifically designed for driving topology reasoning. We hope the reviewer can appreciate the proposed TJS because this metric can help to close the missing gap in terms of the discrete topology graph evaluation. Also, we would like to emphasize that the proposed TJS is only a small part of our contribution. Our core contribution is that we introduce a model-agnostic topology refinement module that helps to produce a better discrete topology graph, which can benefit real-world downstream tasks, like planning and simulation.

---

### Official Review · Reviewer_7aCZ · 2025-10-27

**Soundness:** 2
**Presentation:** 2
**Contribution:** 1
**Rating:** 2
**Confidence:** 4

**Summary:**

This paper proposes a TopoRefine method, a post-training approach that perturbs node features in a discrete graph to introduce noise into its connectivity structure. The model then leverages a Graph Neural Network (GNN) to perform denoising, thereby enhancing the model’s capability in topology reasoning. In addition, the paper introduces the Topology Jaccard Score (TJS), a new metric designed to quantitatively evaluate the structural quality of discrete graphs.

**Strengths:**

1. The method features a simple architecture and incurs low training cost.
2. This is a post-training method that requires no modification to the baseline architecture.
3. The method is integrated with multiple models to validate its effectiveness.

**Weaknesses:**

1. OpenLane-V2 consists of two subsets, Subset A and Subset B, but the paper lacks validation on Subset B.
2. The authors claim that TopoRefine requires only 1.5 hours of training, whereas methods such as SMART take around two days. However, this comparison is unfair because TopoRefine is not trained from scratch. Moreover, existing methods are typically trained on eight GPUs rather than a single one, allowing them to complete training within half a day.
3. The Topology Jaccard Score (TJS) and Topology mAP Score (TOP) both evaluate connectivity reasoning in autonomous driving but from different perspectives. There is no absolute superiority between the two; rather, each captures complementary aspects of topological reasoning. TJS is a discrete, detection-aware metric that measures the overlap between predicted and ground-truth edges, focusing solely on whether topological connections are correct. It is simple, efficient, and interpretable, making it ideal for benchmarking discrete topology reasoning or binary connection prediction. However, TJS overlooks confidence ranking and may not fully reflect the quality of probabilistic predictions. In contrast, TOP, adapted from link prediction in graph learning, assesses how well high-confidence edges correspond to true connections using mean average precision. It captures ranking quality and overall connectivity confidence but is more computationally expensive and sensitive to noisy predictions. In practice, TJS provides a fast and stable measure of structural correctness, while TOP offers a more fine-grained evaluation of model confidence and link reliability.
4. Presenting improvements in percentages is not very intuitive and may come across as overstated.
5. The baseline methods were not originally designed or trained to optimize TJS metric, which explains their weaker performance. Therefore, the large improvement reported on TJS is not entirely convincing. It would be more reliable to demonstrate the improvement through end-to-end (E2E) training that incorporates the TopoRefine method from the beginning.
6. The proposed method involves a large number of hyperparameters, but only partial ablation studies are provided, leaving uncertainty about its sensitivity and generalizability. Moreover, Table 7 evaluates the method on the weakest baseline, whose detection performance is already poor, thereby severely limiting the upper bound of the topology-related metrics.
7. The current method’s topological prediction remains heavily constrained by detection performance. Since TopoRefine is a post-training refinement approach, it inherently inherits the limitations of the underlying detection quality.

**Questions:**

1. Similar to Topologic, integrate the proposed topological refinement into the end-to-end training pipeline, training the model from scratch to demonstrate improvements in both detection and topology performance.
2. Revise the presentation of improvements by showing the absolute difference from the baseline rather than using percentages.
3. Conduct hyperparameter validation based on the end-to-end training framework.

---

> ### Author Response · Authors · 2025-11-24
> **Reviewer 7aCZ - Response [1/9]**
>
> Comment:
> > OpenLane-V2 consists of two subsets, Subset A and Subset B, but the paper lacks validation on Subset B.
>
> We thank the reviewer for pointing this out. Following the suggestion, we conducted additional experiments on OpenLane-V2 Subset B. Since only TopoNet publicly provides pretrained checkpoints for Subset B, we evaluate our refinement module using this model to ensure a fair and consistent comparison. The results (**Table 14 in Appendix B.3**) show that TopoRefine substantially improves both continuous (TOP) and discrete (TJS) metrics on Subset B, consistent with the gains observed on Subset A. This confirms that the refinement module generalizes well across dataset subsets. The results are shown below and the manuscript has been updated accordingly.
>
> ### Results on Subset B
>
> | Method                 | TOP_ll | TOP_lt | TJS_ll | TJS_te |
> |------------------------|-------:|-------:|-------:|-------:|
> | TopoNet (w/o Refine)   |   6.7  |  16.7  |   9.0  |  35.6  |
> | TopoNet (w/ Refine)    |  19.5  |  17.8  |  42.3  |  56.8  |
> | TopoNet Upper Bound    |  21.2  |  20.3  |  48.5  |  66.7  |

---

> ### Author Response · Authors · 2025-11-24
> **Reviewer 7aCZ - Response [2/9]**
>
> Comment:
> > The authors claim that TopoRefine requires only 1.5 hours of training, whereas methods such as SMART take around two days. However, this comparison is unfair because TopoRefine is not trained from scratch. Moreover, existing methods are typically trained on eight GPUs rather than a single one, allowing them to complete training within half a day.
>
> We thank the reviewer for pointing this out. We agree that directly comparing our 1.5-hour training time with methods that train full topology-reasoning models from scratch (e.g., SMART) is not entirely fair. TopoRefine is not a standalone topology reasoning model; it is a lightweight refinement module designed to be added on top of existing models without requiring extra retraining of the backbone. Our goal is to show that this small amount of additional training—only ~1.5 hours—can bring large performance gains when plugged into any existing model.
>
> We do not intend to claim that TopoRefine “trains faster” than full topology models. Instead, our point is that TopoRefine introduces very low additional cost while delivering substantial improvements, making it easy to integrate into current pipelines. **To avoid misunderstanding, we have revised the discussion in the manuscript. See Section 5.2 Feature Extraction and Training.**
>
> These are the contents we add:
> > Training TopoRefine takes about 1.5 hours and validation about 45 minutes on a single H200 GPU on Subset A. Unlike topology reasoning models such as SMART and TopoNet, TopoRefine does not train a full model from scratch; it is a lightweight refinement module that adds only a small amount of extra computation on top of existing models.

---

> ### Author Response · Authors · 2025-11-24
> **Reviewer 7aCZ - Response [3/9]**
>
> Comment:
> > The Topology Jaccard Score (TJS) and Topology mAP Score (TOP) both evaluate connectivity reasoning in autonomous driving but from different perspectives. There is no absolute superiority between the two; rather, each captures complementary aspects of topological reasoning. TJS is a discrete, detection-aware metric that measures the overlap between predicted and ground-truth edges, focusing solely on whether topological connections are correct. It is simple, efficient, and interpretable, making it ideal for benchmarking discrete topology reasoning or binary connection prediction. However, TJS overlooks confidence ranking and may not fully reflect the quality of probabilistic predictions. In contrast, TOP, adapted from link prediction in graph learning, assesses how well high-confidence edges correspond to true connections using mean average precision. It captures ranking quality and overall connectivity confidence but is more computationally expensive and sensitive to noisy predictions. In practice, TJS provides a fast and stable measure of structural correctness, while TOP offers a more fine-grained evaluation of model confidence and link reliability.
>
> We thank the reviewer for the detailed comparison. We agree with the overall point that TJS and TOP measure complementary aspects of topological reasoning. We also clarify two potential misunderstandings.
>
> First, TJS does not overlook confidence ranking. In our evaluation, we compute TJS on the intersection between high-confidence detections and their corresponding high-confidence link predictions. Thus, the ranking signal is still reflected through the confidence thresholds used to determine which links are considered. TJS simply evaluates discrete correctness after thresholding, which is by design since it focuses on binary structural consistency.
>
> Second, regarding probabilistic predictions: TJS is indeed intended for discrete graph correctness rather than probabilistic scoring. This is consistent with its purpose and aligns with TOP, which is specifically designed to capture confidence ranking via mAP. As stated in our paper (line 332), **we treat TJS and TOP as complementary metrics, not substitutes**, and we do not claim that TJS should replace TOP.
>
> We appreciate the reviewer’s discussion, which actually highlights the motivation behind our work: existing metrics emphasize different aspects, and our method can help close the gap between discrete structural correctness and confidence-aware evaluation.

---

> ### Author Response · Authors · 2025-11-24
> **Reviewer 7aCZ - Response [4/9]**
>
> Comment:
> > Presenting improvements in percentages is not very intuitive and may come across as overstated.
>
> We thank the reviewer for the comment. Our intention in reporting percentages was simply to show the relative improvement compared with the baseline, rather than to overstate the gains. To avoid any potential misunderstanding, we now primarily present the absolute performance differences in the main paper and move the percentage changes to the appendix (as a supplementary reference). We believe this makes the results more intuitive while still allowing readers to see the relative scale of improvement if desired. **We have updated the manuscript accordingly in Appendix B.1.**
>
> ### Evaluation across baselines using absolute difference
> We report the detection-conditioned upper bound (UB), the gap to this bound (margin; smaller is better), and discrete graph quality measured by TJS (larger is better). “After’’ results include absolute changes relative to “Before’’ values.
>
> | Method          | UB TOP_ll | UB TOP_lt | margin_ll (Before) | margin_ll (After) | margin_lt (Before) | margin_lt (After) | TJS_ll  Before | TJS_ll After | TJS_lt  Before | TJS_lt After |
> |-----------------|-----------|-----------|---------------------|--------------------|----------------------|---------------------|--------------------|--------------------|---------------------|----------------------|
> | **TopoNet**     | 24.4 | 28.8 | 13.4 | 2.5 (–10.9) | 7.3 | 3.0 (–4.3) | 16.3 | 32.9 (+16.6) | 31.0 | 55.8 (+24.8) |
> | **TopoMLP**     | 25.0 | 31.4 | 3.3  | 0.7 (–2.6) | 4.4 | 2.7 (–1.7) | 18.5 | 59.7 (+41.2) | 31.5 | 59.8 (+28.3) |
> | **Topo2D**      | 25.0 | 29.8 | 2.8  | 0.7 (–2.1) | 3.6 | 2.5 (–1.1) | 18.9 | 55.0 (+36.1) | 34.5 | 51.5 (+17.0) |
> | **TopoLogic**   | 26.0 | 30.6 | 2.0  | 1.5 (–0.5) | 5.3 | 3.3 (–2.0) | 21.0 | 43.9 (+22.9) | 32.5 | 53.3 (+20.8) |
> | **SMART (TopoMLP)** | 40.9 | 38.9 | 3.9 | 0.8 (–3.1) | 5.9 | 3.5 (–2.4) | 38.1 | 87.7 (+49.6) | 36.6 | 82.8 (+46.2) |

---

> ### Author Response · Authors · 2025-11-24
> **Reviewer 7aCZ - Response [5/9]**
>
> Comment:
> > The baseline methods were not originally designed or trained to optimize TJS metric, which explains their weaker performance. Therefore, the large improvement reported on TJS is not entirely convincing. It would be more reliable to demonstrate the improvement through end-to-end (E2E) training that incorporates the TopoRefine method from the beginning.
>
> We thank the reviewer for the insightful comment. Regarding the concern that the improvement on TJS may be less convincing because baseline models were not trained to optimize this metric, **we note that our gains are not limited to TJS**. TopoRefine also brings consistent improvements on the original OpenLane-V2 metrics, including TOP_ll and TOP_lt, which does evaluate confidence ranking and is the primary metric used in prior work. This shows that the improvement is not merely an artifact of TJS but reflects a genuine enhancement in the model’s topological reasoning quality.
>
> On the suggestion of training the entire system end-to-end with TopoRefine from the beginning: the core motivation of our work is that existing models are not designed to optimize discrete graph quality, and we aim to provide a general, lightweight module that can improve their structural correctness without modifying or retraining the base models. Doing full end-to-end retraining would require altering each model’s architecture or loss function to incorporate discrete-graph objectives, which defeats the purpose of a plug-and-play refinement module. Moreover, if we did not change the base model but still retrained it from scratch, the result would be essentially equivalent to using the existing pretrained checkpoint—since **our module is inherently post-training / post-hoc and does not rely on end-to-end training.**
>
> A key advantage of the post-hoc design is practical deployment: TopoRefine can be trained once and applied to many different models and checkpoints without requiring any end-to-end retraining. This flexibility would be lost in an end-to-end setup, which would need a separate full training run for each model and configuration, making it impractical and outside the scope of our goal.
>
> We have added clarifications in the revision to highlight that our contribution is a general, model-agnostic refinement module, not an end-to-end topology reasoning framework. We have added more description in Page 2 Introduction and Page 4 training phase.

---

> ### Author Response · Authors · 2025-11-24
> **Reviewer 7aCZ - Response [6/9]**
>
> Comment:
> > The proposed method involves a large number of hyperparameters, but only partial ablation studies are provided, leaving uncertainty about its sensitivity and generalizability. Moreover, Table 7 evaluates the method on the weakest baseline, whose detection performance is already poor, thereby severely limiting the upper bound of the topology-related metrics.
>
> We thank the reviewer for the helpful comments. We clarify that our method contains very few tunable hyperparameters—the primary one is the SSL loss weight. To address the concern about sensitivity and generalizability, we have added a dedicated sensitivity analysis (**Table 8 in Appendix A.6**), where we vary the SSL weight and observe stable performance across a broad range of settings. This confirms that the refinement behavior is not sensitive to hyperparameter choices.
>
> Regarding Table 7, we agree that the baseline used there has relatively weak detection performance, which naturally constrains its achievable TOP scores. **However, our goal was to illustrate the effect of our refinement module rather than compare absolute detection quality.** Following the reviewer’s suggestion, we conducted additional experiments on stronger baselines—specifically SMART—and found that TopoRefine continues to deliver consistent improvements even when applied on top of high-performance detectors and feature backbones. We also added an ablation using a much weaker ResNet-50 embedding ( **see experiments in Reviewer VE9m - Response [3/7] or Appendix B.5 in the manuscript**), and the refinement gains remain consistent there as well. These results demonstrate that our method does not rely on DINOv2 or the strength of any specific backbone.
>
> Importantly, detection performance itself is not the focus of our work. While better detection indirectly improves TOP metrics, our contribution concerns the refinement of topology once node detections already exist. TopoRefine operates strictly post-hoc: we do not modify or retrain the base topology model, and DINOv2 is used only to construct embedding features for the refinement GNN—not inserted into TopoNet’s end-to-end training. The newly added experiments further reinforce this model-agnostic, post-hoc design.
>
> Overall, our expanded ablation study now covers SSL-weight sensitivity, backbone strength, and performance on stronger baselines. These additional results confirm the robustness and generality of our refinement module. The revised manuscript has been updated accordingly.
>
> ### Performance of different DINOv2 embedding variants applied to SMART
> _All models are evaluated using their tuned best SSL weights._
> | DINOv2 Variant | w_ll | w_lt | TOP_ll | TOP_lt |
> |----------------|------|------|--------|--------|
> | **ViT-G**      | 0.8  | 0.6  | 40.02  | 35.51  |
> | **ViT-L**      | 0.8  | 0.6  | 40.08  | 35.40  |
> | **ViT-B**      | 0.8  | 0.5  | 40.07  | 35.40  |
> | **ViT-S**      | 0.8  | 0.1  | 40.00  | 32.85  |

---

> ### Author Response · Authors · 2025-11-24
> **Reviewer 7aCZ - Response [7/9]**
>
> Comment:
> > The current method’s topological prediction remains heavily constrained by detection performance. Since TopoRefine is a post-training refinement approach, it inherently inherits the limitations of the underlying detection quality.
>
> We thank the reviewer for the observation. We agree that topological prediction quality is influenced by the underlying detection performance. This dependency is natural because all topology models—including ours and existing baselines—operate on the detected nodes provided by the upstream perception module. You can never predict correct topology under incorrect detection.
>
> However, the goal of TopoRefine is not to improve detection itself, but to answer a different question: given the detections a model already produces, how can we obtain better topological predictions from them? As illustrated in the paper (Fig. 1), even with reasonably good detections, topology quality can still be suboptimal, and improving detection alone does not guarantee correct connectivity. Our focus is therefore on refining topological reasoning conditioned on the existing detection outputs.
>
> Importantly, TopoRefine is a post-training refinement module by design. TopoRefine is a totally separate module from the topology reasoning models. This allows it to be plugged into different models without retraining and to bring consistent improvements regardless of the backbone’s detection quality. Better detection modules can further boost performance, but improving detection is outside the scope of this work.
>
> **We have clarified this point in the introduction of the revised manuscript.**

---

> ### Author Response · Authors · 2025-11-24
> **Reviewer 7aCZ - Response [8/9]**
>
> Questions:
> > Similar to Topologic, integrate the proposed topological refinement into the end-to-end training pipeline, training the model from scratch to demonstrate improvements in both detection and topology performance.
> > Conduct hyperparameter validation based on the end-to-end training framework.
>
> We thank the reviewer for the suggestions. We clarify that the goal of our work is fundamentally different from designing an end-to-end topology model such as Topologic. TopoRefine is a post-hoc refinement module that operates after a model produces its detections and initial topology. This design is intentional: it allows our method to work with any existing model, regardless of whether it is image-based, BEV-based, segmentation-based, or uses SD-maps. Because it does not require modifying the backbone or re-training the full pipeline, TopoRefine is truly plug-and-play and can be applied across diverse settings.
>
> Regarding the suggestion to integrate our method into an end-to-end pipeline and train from scratch: doing so would fundamentally change the nature of our approach. End-to-end training would require altering the backbone architecture or loss functions to incorporate discrete-graph objectives, and each new model would need a separate full training run. This would eliminate the key advantages of our method—flexibility, generality, and reusability across different detectors and topology models. In contrast, our post-hoc design allows one module to be trained once and then applied broadly without retraining the base model.
>
> For hyperparameter validation, the same reasoning applies. Since TopoRefine is not part of the end-to-end pipeline, its hyperparameters do not interact with detection training, and end-to-end hyperparameter tuning would not reflect how the method is intended or expected to be used. Instead, we provide sensitivity studies within the post-hoc framework (Appendix B.2), which is the appropriate setting for validating our method.
>
> **We have added clarifications in the introduction and method sections in our revised manuscript to highlight the motivation and benefits of the post-hoc design.**

---

> ### Author Response · Authors · 2025-11-24
> **Reviewer 7aCZ - Response [9/9]**
>
> Question:
> >Revise the presentation of improvements by showing the absolute difference from the baseline rather than using percentages.
>
> We thank the reviewer for the suggestion. We agree that absolute differences provide a more direct and intuitive comparison. Following this feedback, we now present the improvements in terms of absolute performance gains in the main results, and move the percentage changes to the appendix for reference. We believe this revised presentation is clearer while still allowing readers to understand the relative scale of improvements. The manuscript has been updated accordingly in **Table 10 of Appendix B.1.**
>
> ### Evaluation across baselines using absolute values
> We report the detection-conditioned upper bound (UB), the gap to this bound (margin; smaller is better), and discrete graph quality measured by TJS (larger is better). “After’’ results include absolute changes relative to “Before’’ values.
>
> | Method          | UB TOP_ll | UB TOP_lt | margin_ll (Before) | margin_ll (After) | margin_lt (Before) | margin_lt (After) | TJS_ll Before | TJS_ll  After | TJS_lt Before | TJS_lt  After |
> |-----------------|-----------|-----------|---------------------|--------------------|----------------------|---------------------|--------------------|--------------------|---------------------|----------------------|
> | **TopoNet**     | 24.4 | 28.8 | 13.4 | 2.5 (–10.9) | 7.3 | 3.0 (–4.3) | 16.3 | 32.9 (+16.6) | 31.0 | 55.8 (+24.8) |
> | **TopoMLP**     | 25.0 | 31.4 | 3.3  | 0.7 (–2.6) | 4.4 | 2.7 (–1.7) | 18.5 | 59.7 (+41.2) | 31.5 | 59.8 (+28.3) |
> | **Topo2D**      | 25.0 | 29.8 | 2.8  | 0.7 (–2.1) | 3.6 | 2.5 (–1.1) | 18.9 | 55.0 (+36.1) | 34.5 | 51.5 (+17.0) |
> | **TopoLogic**   | 26.0 | 30.6 | 2.0  | 1.5 (–0.5) | 5.3 | 3.3 (–2.0) | 21.0 | 43.9 (+22.9) | 32.5 | 53.3 (+20.8) |
> | **SMART (TopoMLP)** | 40.9 | 38.9 | 3.9 | 0.8 (–3.1) | 5.9 | 3.5 (–2.4) | 38.1 | 87.7 (+49.6) | 36.6 | 82.8 (+46.2) |

---

> > ### Comment · Reviewer_7aCZ · 2025-11-24
> > **Response to authors**
> >
> > Thank you for the clarifications provided in the revision. However, several concerns still remain:
> >
> > 1. Tables 1 and 2 in the main text still report percentage-based results. It is unclear why the authors do not uniformly present relative differences throughout the entire paper.
> >
> > 2. The performance of sub-B is not compared against any state-of-the-art (SOTA) methods.
> >
> > 3. I do not understand the purpose or significance of the “upper bound” experiment. What does this experiment aim to demonstrate?
> >
> > 4. End-to-end training results are necessary to make the claims convincing. A small post-hoc improvement on topology reasoning alone cannot directly influence the design of existing models in the community. The authors do not need to perform end-to-end experiments on all models, but should at least evaluate one strong SOTA baseline.

---

> > > ### Author Response · Authors · 2025-11-30
> > > **R1: Rationale for Reporting Percentage Improvements in the Main Paper**
> > >
> > > > Tables 1 and 2 in the main text still report percentage-based results. It is unclear why the authors do not uniformly present relative differences throughout the entire paper.
> > >
> > > Thank you for the feedback. Our refinement modules build directly on existing topology-reasoning models. Although absolute gains can be reported, percentage improvements are more informative because they normalize results to each baseline’s scale. This makes comparisons fairer, avoids misleading impressions from raw differences, and allows consistent comparison across metrics whose magnitudes vary. Therefore, we report percentage improvements in the main paper and provide absolute differences in the appendix for completeness.

---

> > > ### Author Response · Authors · 2025-11-30
> > > **R2: Clarifying the Choice of Baselines on Subset B Under Reproducibility Constraints**
> > >
> > > > The performance of sub-B is not compared against any state-of-the-art (SOTA) methods.
> > >
> > > Due to reproducibility considerations, we only evaluate topology reasoning models that provide publicly available checkpoints on Subset B; currently, only TopoNet meets this requirement. Our method has already demonstrated consistent improvements over all existing topology-reasoning models on Subset A, as well as over TopoNet on Subset B. These results sufficiently establish the performance gains of our approach. Consequently, adding more experiments on Subset B—where other baselines are not reproducible—would not provide additional meaningful insight.

---

### Official Review · Reviewer_2Uto · 2025-10-28

**Soundness:** 3
**Presentation:** 3
**Contribution:** 3
**Rating:** 6
**Confidence:** 2

**Summary:**

This paper proposes TopoRefine, which addresses the gap between continuous topology prediction, threshold-based discretization, and downstream use in autonomous driving by directly optimizing discrete graph quality. As a post-hoc, plug-and-play module, it operates on the nodes and edges produced by any topology baseline via: (i) self-supervised graph augmentation to construct hard negatives and realistic perturbation distributions; (ii) a lightweight GNN to re-estimate edge existence with relation-aware adaptive weighting; and (iii) fusion/calibration with the baseline scores to yield more reliable discrete connectivity. The paper also introduces the Topology Jaccard Score (TJS), a discrete evaluation metric tailored to connectivity after thresholding. Experiments on an OpenLane-V2 subset across multiple representative baselines show significant improvements on both continuous and discrete metrics, with low overhead and strong generalization.

**Strengths:**

S1. Centers discrete topology quality as both an optimization target and an evaluation focus, filling a gap in settings dominated by continuous scores (e.g., TOP); proposes TJS as a direct measure of discrete connectivity.

S2. Demonstrates stable gains in a plug-and-play manner with unified weights across diverse baselines, along with comprehensive ablations (loss, perturbation, feature interventions) and efficiency reporting.

S3. Improvements in discrete connectivity directly benefit planning/control/map maintenance downstream.

**Weaknesses:**

W1. TJS currently depends on detection matching and thresholding, and may be affected by threshold/matching radius choices and baseline score calibration.

W2. Although the method claims to work with limited/no labels, systematic studies on cross-city/sensor/weather/time-of-day transfer and low-label regimes are missing.

W3. The breakdown of inference latency (feature extraction / refinement / GNN message passing / fusion), memory, and throughput scaling with graph size is not sufficiently detailed.

**Questions:**

Q1. How sensitive are results to the Gaussian perturbation strength and pseudo-node sampling strategy? Do these introduce a domain gap relative to the true prediction-error distribution?

Q2. Under different matching radii and matching strategies (e.g., one-to-many tolerance, maximum-weight matching), does the relative ranking under TJS remain stable?

---

> ### Author Response · Authors · 2025-11-24
> **Reviewer 2Uto - Response [1/5]**
>
> We sincerely appreciate your valuable time and efforts in reviewing this paper. We thank the thoughtful feedback you provided, which significantly improved the quality of this paper. For the potential concerns you bring up, we would like to answer/address them here.
>
> Comment:
> > W1. TJS currently depends on detection matching and thresholding, and may be affected by threshold/matching radius choices and baseline score calibration.
>
> Thank you for the insightful comment. To assess how detection matching, threshold choices, and baseline score calibration affect TJS, we performed additional sensitivity experiments by varying the detection thresholds, topology thresholds, and matching radii under a consistent evaluation setup. As expected, the absolute values of TOP and TJS shift with different thresholding settings, but across all variations the relative ranking and overall trends remain stable. These results indicate that our conclusions do not depend on any particular threshold or matching configuration. The full analysis below is provided in **Table 14 in Appendix B.2.2** and **has been incorporated into the revised manuscript**.
>
> ### Sensitivity analysis of detection and topology thresholds
> Results use the default setup (DINOv2-ViT-L embeddings + TopoNet).
>
> | Det. Th. | Top. Th. | TOP_ll | TOP_lt | TJS_ll | TJS_te |
> |---------:|---------:|-------:|-------:|--------:|--------:|
> | 0.1 | 0.1 | 33.5 | 25.3 | 42.1 | 60.5 |
> | 0.1 | 0.3 | 35.0 | 26.1 | 39.4 | 68.3 |
> | 0.1 | 0.5 | 33.1 | 24.1 | 38.1 | 36.6 |
> | 0.1 | 0.7 | 27.3 | 22.6 | 34.1 | 21.7 |
> | 0.1 | 0.9 | 14.2 | 21.8 | 21.1 | 14.0 |
> | 0.3 | 0.1 | 31.1 | 19.9 | 42.1 | 58.6 |
> | 0.3 | 0.3 | 32.3 | 20.5 | 39.4 | 68.3 |
> | 0.3 | 0.5 | 30.7 | 18.6 | 38.1 | 36.6 |
> | 0.3 | 0.7 | 25.5 | 17.1 | 34.1 | 21.7 |
> | 0.3 | 0.9 | 13.3 | 16.3 | 21.1 | 14.0 |
> | 0.5 | 0.1 | 27.8 | 16.4 | 45.8 | 22.7 |
> | 0.5 | 0.3 | 28.7 | 16.9 | 41.1 | 28.3 |
> | 0.5 | 0.5 | 27.3 | 15.1 | 39.0 | 36.7 |
> | 0.5 | 0.7 | 23.0 | 13.6 | 34.3 | 21.5 |
> | 0.5 | 0.9 | 12.0 | 12.9 | 20.8 | 13.8 |
> | 0.7 | 0.1 | 23.4 | 12.7 | 42.3 | 29.0 |
> | 0.7 | 0.3 | 24.0 | 13.0 | 37.8 | 29.5 |
> | 0.7 | 0.5 | 22.9 | 11.4 | 35.8 | 25.9 |
> | 0.7 | 0.7 | 19.6 | 10.0 | 31.6 | 21.1 |
> | 0.7 | 0.9 | 10.4 | 9.3  | 19.7 | 13.5 |
> | 0.9 | 0.1 | 15.4 | 7.7 | 32.2 | 8.8 |
> | 0.9 | 0.3 | 16.6 | 7.9 | 29.2 | 7.8 |
> | 0.9 | 0.5 | 16.0 | 6.7 | 27.9 | 6.8 |
> | 0.9 | 0.7 | 14.0 | 6.6 | 25.0 | 6.8 |
> | 0.9 | 0.9 | 8.0  | 5.0 | 16.6 | 5.3 |

---

> ### Author Response · Authors · 2025-11-24
> **Reviewer 2Uto - Response [2/5]**
>
> Comment:
> > W2. Although the method claims to work with limited/no labels, systematic studies on cross-city/sensor/weather/time-of-day transfer and low-label regimes are missing.
>
> We thank the reviewer for bringing up this point. To clarify, our method does not rely on limited labels, nor do we assume a low-label setting. The “labels” we use are not things like time-of-day, weather, or multimodal signals. Instead, they are simple training-derived annotations created directly from the data itself. This seems different from how the reviewer interpreted the term “labels,” and we apologize for the confusion and have clarified this in the revision in **Page 2 line 95 to 99**.
>
> (1) we do not assume a low-label regime,
>
> (2) our labels refer to training-derived augmentation labels, and
>
> (3) cross-domain or low-label transfer is not the goal of this paper.
>
> We appreciate the reviewer’s suggestion and agree that studying domain transfer (e.g., day→night, city→city) is valuable, but it represents a separate research direction rather than a direct extension of our method.
>
> These are the contents we added in the manuscript:
>
> > Here, the “labels’’ used for training refer exclusively to these augmentation-derived indicators (real vs.\ perturbed edges) and do not correspond to external factors such as city, sensor type, weather, or time-of-day, which clarifies that our method does not rely on or assume a low-label regime. These domain-transfer conditions fall outside the scope of our work and are unrelated to the type of labels used within our refinement module.

---

> ### Author Response · Authors · 2025-11-24
> **Reviewer 2Uto - Response [3/5]**
>
> Comment:
> > W3. The breakdown of inference latency (feature extraction / refinement / GNN message passing / fusion), memory, and throughput scaling with graph size is not sufficiently detailed.
>
> We thank the reviewer for this helpful suggestion. In the revised manuscript, we have added a detailed runtime and memory analysis in **Appendix B.4**. We benchmark the full TopoRefine inference pipeline on a representative OpenLane-V2 Subset B scene using TopoNet (200 lanes, 100 traffic elements, ~59k candidate edges), running on a single H200 GPU. This setup reflects realistic graph sizes and stress-tests the system under dense connectivity.
>
> The new analysis provides a full breakdown of inference cost across feature extraction/refinement, GNN message passing, and edge-fusion prediction. As shown below, for each frame, the total overhead of TopoRefine is only **6.43 ms and 0.3 MB**, with ~81% of runtime spent on GNN message passing and negligible memory usage across all components. These results demonstrate that TopoRefine is lightweight, efficient, and scalable with graph size. The full tables are included in **Appendix B.4 of the revised manuscript**.
>
> ### Latency Breakdown
> | Component                       | Latency (ms) | Percentage |
> |--------------------------------|-------------:|-----------:|
> | Feature extraction / refinement | 0.21         | 3.2%       |
> | GNN message passing             | 5.21         | 80.9%      |
> | Fusion (edge prediction)        | 0.64         | 9.9%       |
> | **Total**                       | **6.43**     | **100%**   |
>
> ### Memory Overhead
> | Component                       | Memory Overhead |
> |--------------------------------|-----------------:|
> | Feature extraction / refinement | +0.1 MB          |
> | GNN message passing             | +0.0 MB          |
> | Fusion (edge prediction)        | +0.2 MB          |
> | **Total**                       | **+0.3 MB**      |

---

> ### Author Response · Authors · 2025-11-24
> **Reviewer 2Uto - Response [4/5]**
>
> Comment:
> > Q1. How sensitive are results to the Gaussian perturbation strength and pseudo-node sampling strategy? Do these introduce a domain gap relative to the true prediction-error distribution?
>
> We thank the reviewer for the helpful question. In our method, Gaussian perturbation is simply used as a standard noise-injection strategy for graph augmentation, not as a model of the true prediction-error distribution. Specifically, we add Gaussian noise to bounding-box and polyline coordinates to generate the synthetic pseudo-nodes. Since each lane/polyline or bounding box is described by several to dozens of coordinates, the aggregated perturbation over these coordinates can be well approximated by a Gaussian distribution. This is consistent with the central limit theorem and common practice in augmentation-based training.
>
> The purpose of this noise is therefore to add small, smooth coordinate-level variations around the annotated geometry, rather than to mimic a specific real-world error process. As a result, any domain gap introduced by the perturbation is theoretically limited: the perturbed samples stay close to the original annotations, and the model mainly learns robustness to small geometric changes. Empirically, we also observe that the refinement module is stable across a wide range of perturbation strengths. We tested different Gaussian standard deviations and obtained very similar performance; we include these sensitivity results in **Appendix B.2.1.** We also evaluated alternative pseudo-node sampling strategies and found the results to be consistent.
>
> Overall, both theory and experiments suggest that Gaussian perturbation does not introduce a meaningful domain gap for our framework. We have clarified this point in the revised manuscript.
>
> ### Sensitivity Analysis: Lane Std and TE Std in Gaussian Perturbation
>
> | Lane Std | TE Std | TOP_ll | TOP_lt | TJS_ll | TJS_te |
> |----------|--------|--------|--------|--------|--------|
> | 0.3      | 12     | 21.84  | 25.79  | 32.92  | 55.81  |
> | 0.5      | 12     | 21.84  | 25.76  | 32.92  | 56.32  |
> | 0.7      | 12     | 21.84  | 25.71  | 32.92  | 53.29  |
> | 0.9      | 12     | 21.84  | 25.50  | 32.92  | 51.68  |
> | 0.1      | 12     | 21.84  | 25.74  | 32.92  | 53.49  |
> | 0.1      | 10     | 21.83  | 25.67  | 32.86  | 54.47  |
> | 0.1      | 8      | 21.84  | 25.71  | 32.92  | 53.13  |
> | 0.1      | 6      | 21.83  | 25.63  | 32.92  | 52.67  |
> | 0.1      | 4      | 21.84  | 25.78  | 32.92  | 55.77  |

---

> ### Author Response · Authors · 2025-11-24
> **Reviewer 2Uto - Response [5/5]**
>
> Comment:
> >  Under different matching radii and matching strategies (e.g., one-to-many tolerance, maximum-weight matching), does the relative ranking under TJS remain stable?
>
> We thank the reviewer for the question. TJS must follow the official OpenLane-V2 bipartite matching protocol, since all node correspondences (including pseudo-nodes) are defined through the OL-V2 matching rules. Alternative matching strategies (e.g., one-to-many or maximum-weight matching) would break consistency with the dataset’s supervision and therefore cannot be used.
>
> To address the concern about robustness, we performed a full sensitivity analysis over different detection and topology confidence thresholds. All experiments use our standard setup (DINOv2-ViT-L embeddings + TopoNet) to ensure a consistent evaluation pipeline.
>
> **Findings:**
> • Both TOP and TJS vary with thresholds, since thresholding directly controls graph sparsity (lower thresholds → higher recall → higher metrics; higher thresholds → more pruning → lower metrics).
> • Despite these shifts, the **relative ranking of configurations remains stable** across a wide range of thresholds.
> • This confirms that our evaluation conclusions **do not depend on any specific threshold choice**.
>
> These results and the accompanying discussion are included in **Appendix B.2.2 of the revised manuscript**. The raw values are also shown in **Reviewer 2Uto - Response [1/5]**.

---

> > ### Comment · Reviewer_2Uto · 2025-11-25
> >
> > Thanks for the detailed response. Although I’m not familiar with this area, it has improved my understanding, and I will keep my current rating.

---

### Author Response · Authors · 2025-11-24
**Global Response to the Area Chair**

Thank you for handling our submission and for the constructive feedback from all reviewers. Below, we summarize the core problem addressed by this work, what our method actually does, the contributions, the main findings, and the revisions made in response to reviewer concerns.

**Problem.**
Existing topology reasoning models output a continuous topology graph in which all candidate lane–to–lane and lane–to–traffic connections have continuous confidence scores. However, real-world downstream task like planning and simulation requires a discrete topology graph with discrete edge connections. This creates a big gap between the development of driving topology reasoning methods and their application to downstream tasks in the real world. This gap has either not been addressed/closed in prior work or explicitly examined during the evaluation. Current benchmarks evaluate only the continuous predictions, leaving a gap between high confidence scores and unreliable discrete topology graph(Figures 1–2).

**What we do.**
We identify this missing gap and propose TopoRefine, a lightweight post-hoc, model-agnostic GNN refinement module trained via self-supervised learning. Specifically, this lightweight post-hoc module will refine the continuous topology graph output by the base topology models (e.g. TopoNet and TopoMLP) into a more reliable discrete topology graph. Once trained, it can seamlessly integrate with any different base topology model to further refine its topology graph without extra training effort. Importantly, TopoRefine does not modify or retrain the base topology models, does not use a stronger backbone like DINOv2 to improve detection, and does not need to integrate into any end-to-end pipeline. We take the predictions of an existing base topology model as-is and refine only the connectivity using a separately trained SSL-GNN. This design is central to our motivation and was misunderstood by some reviewers.

**Contributions.**
We formalize the distinction between continuous and discrete topology graphs and motivate why discrete correctness is essential. We introduce TJS, a detection-aware metric designed specifically to evaluate discrete topology connectivity. More importantly, we propose TopoRefine, a plug-and-play refinement module that improves discrete connectivity output by any existing topology reasoning model without modifying or retraining the existing topology reasoning model.

**Key misunderstanding.**
A central misunderstanding across the reviews is the assumption that our improvements arise from using DINOv2 or from enhancing the detection performance of the underlying model. This is not the motivation or mechanism of our approach. DINOv2 is used only to construct feature embeddings for training the refinement module; it is never integrated into the base topology reasoning model, never used to modify detection, and never involved in end-to-end training. TopoRefine is entirely post-hoc and model-agnostic: it refines only the predicted connectivity (using a pre-trained GNN) and leaves all other components of the topology model untouched. We have revised the motivation section to avoid confusion in the updated manuscript.

## Summary of Revisions Added in Response to Reviewers
We made several updates to clarify the motivation, address misunderstandings, and strengthen the empirical evidence.
**1. Clarified continuous vs. discrete graphs**

Added definitions and motivation in Introduction, Page 1 line 49 – Page 2 line 75.

**2. Clarified the meaning of “labels” and removed the low-label misunderstanding**
Updated Page 2 line 90–99.

**3. Added Subset B experiments**
Results added in Appendix B.

**4. Added extensive sensitivity analyses**
– Threshold & matching radius: Appendix B.2.2 (Table 14)
– Gaussian perturbation: Appendix B.2.1
– SSL loss weight & hyperparameters: Appendix A.6

**5. Clarified that TopoRefine is post-hoc and model-agnostic**
Strengthened explanation in Introduction (Page 2) and Training Phase (Page 4) to highlight the key misunderstanding:
we do not improve detection or retrain the base models; we train a separate SSL-GNN and apply it only post-hoc.

**6. Added Subset B baselines and stronger-baseline refinement results**
Clarified in Appendix B.3.

**7. Updated absolute improvements**
Presented absolute gains in the main paper and moved percentage changes to Appendix B.1.

**8. Clarified matching protocol and evaluation setup**
Explicitly stated adherence to official OpenLane-V2 evaluation in Section 5.1.

**9. The breakdown of inference latency**
Clarified in Appendix B.4.

**10. Demonstrated backbone-independence of TopoRefine**

Added ResNet-50 ablation (Appendix B.5) showing that TopoRefine improves all models even with weak embeddings, confirming that our gains do not rely on DINOv2 or detection improvement.

---

### Meta-Review · Area_Chair_VEze · 2025-12-21

**Summary:**

The submission proposes TopoRefine, a post-hoc refinement module designed to improve the discrete connectivity of topology graphs in autonomous driving settings. The authors claim that by processing the continuous outputs of existing topology models with a lightweight, self-supervised Graph Neural Network (GNN), the refined discrete graph better serves downstream tasks such as planning and simulation. Additionally, the paper introduces the Topology Jaccard Score (TJS) as a metric tailored for assessing the accuracy of discrete connectivity. In response to the reviewers' comments, the authors have revised the manuscript to clarify the distinction between continuous and discrete topological graphs, adjusted some experimental details, and added further ablation studies and sensitivity analyses.

**Reviewer Concerns:**

The reviewers raised several concerns regarding both the methodology and the experimental evaluation. One consistent point among the reviews is that the contributions appear marginal, with the improvements being sensitive to threshold choices and detection performance, and the proposed TJS not offering a notionally novel perspective beyond traditional IoU-based metrics. Concerns were also expressed about the experimental design, including the limited evaluation on only one subset of the dataset, the lack of end-to-end integration studies, and insufficient breakdowns of hyperparameter sensitivity and inference performance. Collectively, these issues cast doubt on whether the approach justifies its claims, particularly when weighed against the potential added complexity and the reliance on post-hoc refinement without addressing underlying detection limitations.

**Reviewer Scores:**

Given the mixed nature of the individual reviews—some acknowledged incremental improvements, while others highlighted significant shortcomings in clarity, experimental rigor, and methodological novelty—the overall recommendation for this submission is to reject it. Although addressing the gap between continuous and discrete topology predictions is an essential and relevant problem, the current submission does not offer sufficient innovation or convincing empirical evidence to warrant acceptance at this time.

---

### Decision · Program_Chairs · 2026-01-26

Reject